**Environmental Effects on Aerosol-Cloud Interaction in non-precipitating MBL**
**Clouds over the Eastern North Atlantic**
Xiaojian Zheng[1], Baike Xi[1], Xiquan Dong[1], Peng Wu[2], Timothy Logan[3] and Yuan Wang[4,5]
[1]Department of Hydrology and Atmospheric Sciences, University of Arizona, Tucson, AZ, USA
[2]Pacific Northwest National Laboratory, Richland, WA, USA
[3]Department of Atmospheric Sciences, Texas A&M University, College Station, TX, USA
[4]Division of Geological and Planetary Sciences, California Institute of Technology, Pasadena, CA,
USA
[5]Jet Propulsion Laboratory, California Institute of Technology, Pasadena, CA, USA
**Correspondence**: Baike Xi (baikex@arizona.edu)
**Abstract.** Over the eastern north Atlantic (ENA) ocean, a total of 20 non-precipitating single-layer
marine boundary layer (MBL) stratus and stratocumulus cloud cases are selected to investigate the
impacts of the environmental variables on the aerosol-cloud interaction ($ACI_r$) using the ground-based
measurements from the Department of Energy Atmospheric Radiation Measurement (ARM) facility at
the ENA site during 2016 – 2018. The $ACI_r$ represents the relative change of cloud-droplet effective
radius $r_e$ with respect to the relative change of cloud condensation nuclei (CCN) number concentration
at 0.2% supersaturation ($N_{CCN,0.2\%}$) in the water vapor stratified environment. The $ACI_r$ values vary from
-0.01 to 0.22 with increasing sub-cloud boundary layer precipitable water vapor ($PWV_{BL}$) conditions,
indicating that $r_e$ is more sensitive to the CCN loading under sufficient water vapor supply, owing to the
combined effect of enhanced condensational growth and coalescence processes associated with higher
$N_c$ and $PWV_{BL}$. The principal component analysis shows that the most pronounced pattern during the
selected cases is the co-variations of the MBL conditions characterized by the vertical component of
turbulence kinetic energy ($TKE_w$), decoupling index ($D_i$), and $PWV_{BL}$. The environmental effects on
$ACI_r$ emerge after the data are stratified into different $TKE_w$ regimes. The $ACI_r$ values, under both
relatively lower and higher $PWV_{BL}$ conditions, increase more than double from the low $TKE_w$ to high
$TKE_w$ regime. It can be explained by the fact that stronger boundary layer turbulence maintains a well-
mixed MBL, strengthening the connection between cloud microphysical properties and the below-cloud
CCN and moisture sources. With sufficient water vapor and low CCN loading, the active coalescence

process broadens the cloud droplet size spectra, and consequently results in an enlargement of $r_e$. The enhanced activation of CCN and the cloud droplet condensational growth induced by the higher below-cloud CCN loading can effectively decrease $r_e$, which jointly presents as the increased $ACI_r$. This study examines the importance of environmental effects on the $ACI_r$ assessments and provides observational constraints to future model evaluations on aerosol-cloud interactions.

## 1. Introduction

Clouds are one of the most important parts of the Earth's climate system. They can impact the global climate by modulating the radiative balance in the atmosphere. Moreover, the radiative effects of cloud adjustments due to aerosols remain one of the largest uncertainties in climate modeling (IPCC, 2013). Over the oceanic area, the lower troposphere is dominated by marine boundary layer (MBL) clouds. MBL clouds can persistently reflect the solar radiation by their long-lasting nature maintained by cloud-top radiative cooling, and therefore act as a major modulator of the Earth's radiative budget (Seinfeld et al., 2016). The climatic importance of MBL cloud radiative properties is primarily induced by cloud microphysical properties such as cloud-droplet number concentration ($N_C$) and effective radius ($r_e$), and has been intensively investigated by many researchers (Garrett and Zhao, 2006; Rosenfeld, 2007; Wood et al., 2015; Seinfeld et al., 2016). The ambient aerosol conditions can influence these cloud microphysical properties via the aerosol-cloud interaction (ACI). Compared to the clean regions, clouds under the regions having relatively higher below-cloud aerosol concentrations exhibited smaller cloud droplets (reduced $r_e$ and increased $N_C$) and enhanced both cloud liquid water contents and optical depths (McComiskey et al., 2009; Chen et al., 2014; Wang et al., 2018). The changes of MBL cloud microphysical properties induced by aerosols have been investigated from previous studies using in-situ measurements, ground- and satellite-based observations, and model simulations in multiple oceanic areas such as the eastern Pacific and eastern Atlantic (Twohy et al., 2005; Lu et al., 2007; Hill et al., 2009; Costantino and Bréon, 2010; Mann et al., 2014; Dong et al., 2015; Diamond et al., 2018; Yang et al., 2019; Zhao et al., 2019; Wang et al., 2020).

The assessments of ACI, particularly using ground-based remote sensing, vary in terms of the quantitative values, which represent the different cloud susceptibilities to aerosol loadings. Owing to the numerous approaches in assessing the ACI, such as the spatial and temporal scales, $N_c$ and $r_e$ retrieval methods, and more importantly, the different aerosol proxies used in the ACI quantification, different ACI results could be achieved. For example, the studies using total aerosol number concentration and

aerosol scattering/extinction coefficients to represent the aerosol loadings would result in relatively lower
ACI values (Pandithurai et al., 2009; Liu et al., 2016). This is primarily attributed to the inclusion of
aerosol species with different abilities to activate, which is determined by their physicochemical
properties, and thus will cause non-negligible uncertainties in capturing the information of aerosol
intrusion to the cloud (Feingold et al. 2006; Logan et al., 2014). While some studies found relatively
higher ACI values using cloud condensation nuclei (CCN) number concentration ($N_{CCN}$), presumably
due to the fact that CCN represents the portion of aerosols that can be activated and possesses the
potential ability to further grow into cloud droplets, this favorably yields a more straightforward
assessment of ACI (McComiskey et al., 2009; Qiu et al., 2017; Zheng et al., 2020). It is noteworthy that
the ACI variations have been found to have both increasing and decreasing trends in response to changing
environmental water availability (Martin et al., 2004; Kim et al., 2008; McComiskey et al., 2009;
Pandithurai et al., 2009; Martin et al., 2011; Liu et al., 2016; Zheng et al., 2020). Although these
contradicting results have been postulated due to multiple factors such as cloud adiabaticity,
condensational growth, collision coalescence, and atmospheric thermodynamics and dynamics, the
underlying mechanisms in altering the ACI and causing the uncertainties in the ACI assessments remain
unclear. Therefore, further studies are necessary (Fan et al., 2016; Feingold and McComiskey, 2016;
Seinfeld et al., 2016).
The Eastern North Atlantic (ENA) is a remote oceanic region that features persistent but diverse
subtropical MBL clouds, owing to complex meteorological influences from the semi-permanent Azores
High and prevailing large-scale subsidence (Wood et al., 2015). The ENA has become a favorable region
to study the aerosol indirect effects on MBL clouds under a relatively clean environment with occasional
intrusions of long-range transport of continental air mass (Logan et al., 2014; Wang et al., 2020). The
atmospheric radiation measurement (ARM) program established the ENA permanent observatory site on
the northern edge of Graciosa Island, Azores, in 2013, which continuously provides comprehensive
measurements of the atmosphere, radiation, cloud, and aerosol from ground-based observation
instruments. Owing to the location of the site, which sits in between the boundaries of mid-latitude and
subtropical regimes, the ENA is under the mixed influence of diverse meteorological conditions. In terms
of the aerosol influence on the cloud properties, the roles of meteorological factors on cloud formation
and development are not negligible and hence are being explored in this study. The large-scale
thermodynamic variables of the lower troposphere are widely used, such as the lower tropospheric
stability (LTS), where the higher LTS values are found to be associated with a relatively shallow and
well-mixed marine boundary layer, and are prone to stratiform cloud formations with higher cloud
fractions (Klein and Hartmann, 1993; Wood, 2012; Wood and Bretherton, 2006; Yue et al., 2011;

Rosenfeld et al., 2019), especially over the subtropical ocean such as the northeast Atlantic. Over the ENA site, the spatial gradient of the LTS has been studied to be associated with the contribution terms of MBL turbulence and the wind directional change (Wu et al., 2017).

In the cloud-topped MBL which is maintained by cloud-top radiative cooling, the buoyancy generation and shear contribute most to the turbulence kinetic energy (TKE) production (Nicholls, 1984; Hogan et al., 2009), where the intensity of turbulence denotes the coupling of MBL clouds to the below-cloud boundary layer. In terms of the cloud droplet growth process, especially in a clean environment with low $N_{CCN}$ below the cloud layer, the cloud droplets at the cloud base experience rapid growth via the diffusion of water vapor, and subsequently enter the regime of active coalescence (Rosenfeld and Woodley, 2003; Martins et al., 2011). The intensive turbulence effectively modulates the cloud droplet growth by strengthening the coalescence process and the cloud cycling (Feingold et al., 1996, 1999; Pawlowska et al., 2006). In particular, the unique topography of Graciosa Island induces an island effect which could cause disturbances in the updraft and hence impact the MBL turbulence, depending on the surface wind directions (Zheng et al., 2016). The environmental effects on the MBL cloud formation and development processes and cloud microphysical properties have been widely implemented and considered in climate modeling (Medeiros and Stevens, 2011; West et al., 2014; Zhang et al., 2016). Thus, it is important to provide observational constraints on the environmental effects. The assessment of ACI from the ground-based perspective highly relies on the sensitivities of cloud droplet number concentrations and size distribution to the changing of below-cloud CCN loadings. Hence, studying the relationship between the environmental effect and the MBL cloud microphysical responses is a nontrivial task.

In this study, we target the non-precipitating single-layer MBL stratus and stratocumulus clouds during the period between September 2016 and May 2018 and examine the role of thermodynamical and dynamical variables on ACIs. This study aims to advance the understanding of ACI by disentangling the environmental effects and providing observational constraints on quantifying the ACI when modeling aerosol effects on MBL clouds. The ground-based observations and retrievals, and the reanalysis are introduced in section 2. Section 3 describes the aerosol, cloud and meteorological properties, and the variations of cloud microphysical properties under different environmental regimes. Moreover, the ACIs under given water vapor conditions and the roles of environmental effects on ACI are discussed in Section 3. The conclusion of the key findings and the future work are presented in section 4.

## 2. Data and methods

### 2.1 Cloud and aerosol properties

The cloud boundaries at the ARM ENA site are primarily determined by the ARM Active Remotely-
Sensed Cloud Locations (ARSCL) product, which is a combination of data detected by multiple active
remote-sensing instruments, including the Ka-band ARM Zenith Radar (KAZR) and laser ceilometer.
The KAZR has an operating frequency at 35 GHz and is sensitive in cloud detection with very minimum
attenuation up to the cloud top height (Widener et al., 2012). The temporal and vertical resolutions of
KAZR reflectivity are 4 seconds and 30 m, respectively. The ceilometer operates at 910 nm and its
attenuated backscatter data can be converted to the cloud base height up to 7.7 km with an uncertainty
of ~10 m (Morris, 2016). Combing both KAZR and ceilometer measurements, the cloud base ($z_b$) and
top ($z_t$) heights can be identified accordingly. The single-layer low cloud is defined as having a cloud
top height lower than 3 km, with no additional cloud layer in the atmosphere above (Xi et al., 2010).
The cloud microphysical properties are retrieved from a combination of ground-based observations,
including KAZR, ceilometer, and microwave radiometer. The detailed retrieval methods and procedures
are described in Wu et al. (2020a). The retrieved cloud microphysical properties, both in time series and
vertical profiles, have been validated using the collocated aircraft in-situ measurements during the
Aerosol and Cloud Experiments in the Eastern North Atlantic field campaign (ACE-ENA). The retrieval
uncertainties are estimated to be ~15% for cloud droplet effective radius ($r_e$), ~35% for cloud droplet
number concentration ($N_c$), and ~30% for the cloud liquid water content (LWC) (Wu et al., 2020a).
Furthermore, the cloud adiabaticity is calculated using the retrieved in-cloud vertical profile of LWC and
the adiabatic $\text{LWC}_\text{ad}$. The $\text{LWC}_\text{ad}$ is given by $\text{LWC}_{\text{ad}(z)} = \Gamma_{ad}(z - z_b)$, following the method in Wu et
al. (2020b), where $\Gamma_{ad}$ denotes the linear increase of LWC with height under an ideal adiabatic condition
(Wood, 2005). The cloud adiabaticity ($f_{ad}$) is defined as the ratio of LWC to $\text{LWC}_\text{ad}$.
The surface CCN number concentrations ($N_{CCN}$) are measured by the CCN-100 (single-column)
counter. Since the supersaturation (SS) levels cycle between approximately 0.10% and 1.10% within
one hour, $N_{CCN}$ under a relatively stable supersaturation level has to be carefully calculated to rule out
the impact of supersaturation on $N_{CCN}$. This study adopts the interpolation method given by $\text{N}_{CCN} = c\text{SS}^k$
(Twomey, 1959), where parameters c and k are fitted by a power-law function for every periodic cycle.
In this study, the supersaturation level of 0.2% is used because it represents typical supersaturation
conditions of boundary-layer stratiform clouds (Hudson and Noble, 2013; Logan et al., 2014; Wood et
al., 2015; Siebert et al., 2021), and $N_{CCN}$ at 0.2% supersaturation (hereafter $N_{CCN,0.2\%}$) is interpolated to
a 5-min temporal resolution.

**2.2 Environmental conditions and cloud case selections**

The integrated precipitable water vapor (PWV) is obtained from a 3-channel microwave radiometer

(MWR3C), which operates at three frequency channels of 23.834, 30, and 89 GHz. The uncertainty of
PWV is estimated to be ~0.03 cm (Cadeddu et al., 2013). To capture the information of MBL water vapor
more accurately, the sub-cloud boundary layer integrated precipitable water vapor ($PWV_{BL}$) is calculated
using the interpolated sounding product following:
$PWV_{BL} = \frac{1}{\rho_w} \sum (z_{i+1} - z_i) * (\rho_{v,i+1} + \rho_{v,i})/2,$             (1)
where the $\rho_w$ is the liquid water density and the $\rho_v$ is the water vapor density collected from the
Interpolated Sounding and Gridded Sounding Value-Added Products (Toto and Jensen, 2016), the
subscripts $i$ and $i+1$ represent the bottom and top of each interpolated sounding height layer. Both
PWV and $PWV_{BL}$ are temporally collocated to 5-min resolutions and plotted against each other in Fig.
S1a to test the contribution of $PWV_{BL}$ to PWV. The Pearson correlation coefficient of 0.85 shows that
the $PWV_{BL}$ are strongly positively correlated with PWV, while the distribution of the percentage ratio of
$PWV_{BL}$ to PWV (Fig. S1b) indicates that, on average, $PWV_{BL}$ contributes to ~58% of PWV. Considering
the cloud-topped MBL, the majority of cases (~74%) associate with a relatively moist boundary layer
compared to the amount of water vapor in the free troposphere, where $PWV_{BL}$ already contributed over
50% of the total column PWV. In contrast, only ~9% of cloud samples occur under a relatively dry
boundary layer and moist free troposphere, where $PWV_{BL}$ contributions are less than 40%. In general,
PWV can well capture the variation of $PWV_{BL}$. In the rest of the study, $PWV_{BL}$ are used, as it represents
the sub-cloud boundary layer water vapor availability which is more closely related to the MBL cloud
processes.

The LTS parameter is used as a proxy of large-scale thermodynamic structure and is defined as the

difference between the potential temperature at 700 hPa and surface ($\theta_{700} - \theta_{sfc}$). The LTS values are
calculated from European Centre for Medium-Range Weather Forecasts (ECMWF) model outputs of
potential temperature, by averaging over a grid box of 0.56°×0.56° centered at the ENA site. To match
the temporal resolutions of the other variables, the original 1-hour LTS data are downscaled to 5-min
under the assumption that the large-scale forcing would not have significant changes within an hour.

The boundary layer decoupling condition is represented by the decoupling index ($D_i$), which is

given by $D_i = (z_b - z_{LCL})/z_b$, where the $z_{LCL}$ is the lifting condensation level calculated analytically
following the method in Romps (2017), with an uncertainty of around 5 m. The surface temperature,
pressure, relative humidity, and mass fraction of water vapor are used in the $z_{LCL}$ calculation, as long as
the vector-averaged wind directions (in 360° coordinate) over the ENA site are obtained from the ARM
surface meteorology systems (ARM MET handbook, 2011).
As for the boundary layer dynamics, the higher-order moments of vertical velocity are widely used
in different model parameterization practices, such as higher-order turbulence closure and probability
density function methods (Lappen and Randall, 2001; Zhu and Zuidema, 2009; Ghate et al., 2010). The
vertical velocity variance can be used to represent the turbulence intensity in the below-cloud boundary
layer (Feingold et al., 1999). In this study, the vertical component of the turbulence kinetic energy ($TKE_w$)
is used, which is defined as:
$TKE_w = \frac{1}{2}\overline{(w')^2}$ ,                           (2)
where the $(w')^2$ is the variance of vertical velocity measured from the Doppler lidar standard 10-min
integration, which is collected in the Doppler Lidar Vertical Velocity Statistics Value-Added Product
(Newson et al., 2019). The noise correction has been applied to reduce the uncertainty of the variance to
~10% (Hogan et al., 2009; Pearson et al., 2009). In this study, the mean value of $TKE_w$ in the sub-cloud
boundary layer proportion of the Doppler lidar range is used, and the data temporal resolution is further
downscaled to 5-min for temporal collocation purposes.
In this study, the non-precipitating cloud periods are determined when the KAZR reflectivity at the
ceilometer-detected cloud base height range does not exceed -37 dBZ (Wu et al., 2015, 2020b), which
extensively rules out the wet-scavenging depletion on below-cloud CCN (Wood, 2006) and ensures the
accuracy in capturing the below-cloud CCN loadings. Both retrieved cloud microphysical properties and
CCN data are available from September 2016 to May 2018 and confine this period in this study.

**3.  Result and Discussion**
**3.1 Aerosol, cloud, and meteorological properties of selected cloud cases**
A total of 20 non-precipitating cloud cases are selected in this study, with the detailed time periods
listed in Table 1, including 1143 samples with temporal resolutions of 5-min, which corresponds to ~95
hours. Among the selected cases, there are three, eight, five, and four cases for Spring, Summer, Fall,
and Winter seasons, respectively. MBL clouds often produce precipitation in the form of drizzle (Wood
2012, Wu et al., 2015, 2020b). A recent study of the seasonal variation of the drizzling frequencies (Wu
et al., 2020b) showed that the MBL clouds in the cold months (Oct-Mar) have the highest drizzling
frequency of the year (~70%), while the clouds in the warm months (Apr-Sept) are found to have a lower
chance of drizzling (~45%). Therefore, the selection of a non-precipitating single-layer low cloud case
that lasts at least 2 hours is limited, with only 6 cases found in the cold months and 14 cases found during
the warm months.
The probability distribution functions (PDFs) of the aerosol and cloud properties, and the
environmental conditions for the selected cases are shown in Fig. 1. The PDF of $N_{CCN,0.2\%}$ presents a
normal distribution with a mean value of 215 cm$^{-3}$ and median value of 217 cm$^{-3}$. About 97% of the
$N_{CCN,0.2\%}$ samples lie below 350 cm$^{-3}$ and represents a relatively clean environment (Logan et al., 2014,
2018). A few instances of aerosol intrusions (~3%) with higher $N_{CCN,0.2\%}$ were likely a result of
continental air mass transport from North America, Europe, and Africa (Logan et al., 2014; Wang et al.,
2020). As for the cloud microphysical properties, the cloud-layer mean $N_c$ and $r_e$ (Fig. 1b and 1c) are
also both normally distributed with median values close to the mean values. The majority of the $N_c$
values (~91%) are lower than 125 cm$^{-3}$ with a mean value of 86 cm$^{-3}$, and the $r_e$ distribution peaks
between 9 - 11 µm with a mean value of 10.1 µm. Both $N_c$ and $r_e$ values fall in the typical ranges of the
non-precipitating MBL cloud characteristics over the ENA site (Dong et al., 2014; Wu et al., 2020b).
The distribution of $f_{ad}$ is slightly skewed to the left with a median value of 0.66 (Fig. 1d), indicating that
the bulk of cloud samples are close to adiabatic environments, while the left tail denotes a wide range of
cloud sub-adiabaticities, which allows us to investigate the role of cloud adiabaticities on the cloud
microphysical variations.
For all selected cases, the LTS, which represents the large-scale thermodynamic structure, is
distributed bimodally across the range from 14K to 23K with mean and median values of 19.1K in Fig.
1e. A higher LTS magnitude represents a relatively stable environment and is favorable to the formation
of marine stratocumulus (Medeiros and Stevens, 2011; Gryspeerdt et al., 2016). Note that the median
LTS of 19.1 K in this study is close to the separation threshold of 18.55K suggested by prior studies to
distinguish the marine stratocumulus from a global assessment of marine shallow cumulus clouds
(Smalley and Rapp, 2020). Therefore, leveraging the demarcation line at 19.1K may allow us to
investigate the aerosol-cloud relationships under contrasting thermodynamic regimes. The PDF of $D_i$
parameter spreads widely with a median value of 0.34 for the selected cases (Fig. 1f), which provides an
opportunity to study the cloud sample behaviors under MBL conditions range from well-mixed to
decoupled. Higher $D_i$ values indicate more decoupled MBL with weaker turbulence which cannot
sufficiently maintain the well-mixed MBL, while lower $D_i$ values often associate with stronger
turbulence which maintains a coupled MBL (Jones et al., 2011). As an indicator of the below-cloud
boundary layer turbulence, the TKE$_w$ values present a gamma distribution that is highly skewed to the
right (Fig. 1e), with a mean value of 0.11 and a median value of 0.08 m$^2$s$^{-2}$. About half of the cloud
samples are observed within a relatively less turbulent environment (which is also implied by the higher
half of $D_i$), suggesting weak connections between the cloud layer and the below-cloud boundary layer.
The other half of the cloud samples, with relatively higher $TKE_w$ values up to $0.4 \ m^2/s^2$, imply tighter
connections between cloud microphysical properties and below-cloud boundary layer accompanied by
intensive turbulent conditions, which is favorable to enhance cloud droplet growth (Albrecht et al., 1995;
Hogan et al., 2009; Ghate et al., 2010; West et al., 2014; Ghate and Cadeddu, 2019).

It is noteworthy that $PWV_{BL}$ values exhibit a bimodal distribution with a median value of 1.2 cm

(Fig. 1f). About 49% of the samples have their $PWV_{BL}$ values in the range of 0.4 - 1.2 cm with the first
peak in 0.6 - 0.8 cm, and 51% of the samples have $PWV_{BL}$ values higher than 1.2 cm with a second peak
in 1.6 - 1.8 cm, which may be due to the seasonal difference of the selected cases. Fig. S2 shows the
seasonal variation of the $PWV_{BL}$ from 2016 to 2018 when single-layered low clouds are present. The
monthly $PWV_{BL}$ values are as low as ~ 0.9 cm and remain nearly invariant from January through March,
then increase to ~ 2.0 cm (doubled) in September, and decrease dramatically to the winter months.  The
selected cloud cases are distributed across the seasons, with ~34% of the samples occurring during the
months with the lowest mean $PWV_{BL}$ (Jan-Mar), while ~43% of the samples fall in the highest $PWV_{BL}$
months (Jun-Sept). These two different $PWV_{BL}$ regions will provide a great opportunity for us to further
examine the ACI under relatively lower and higher water vapor conditions.

**3.2 Dependent of cloud microphysical properties on CCN and $PWV_{BL}$**

Figure 2 shows the cloud microphysical properties as a function of $N_{CCN,0.2\%}$ and $PWV_{BL}$ for the

samples from 20 selected cases. As illustrated in Fig. 2a, there is a statistically significant positive
correlation ($R^2$=0.9) between $ln(N_c)$ and $\ln(N_{CCN,0.2\%})$. The linear fit of $ln(N_c)$ to $\ln(N_{CCN,0.2\%})$ is then
mathematically transformed to a power-law fitting function of $N_c$ to $N_{CCN,0.2\%}$, and plotted as dash lines
in Fig. 2a. The power-law fitting indicates that 90.3% of the variation in binned $ln(N_c)$ can be explained
by the change in the binned $ln(N_{CCN,0.2\%})$ and further suggests that with more available below-cloud
CCN, higher number concentrations are expected. The logarithmic ratio $\partial ln(N_c)/\partial ln(N_{CCN,0.2\%})$ is
computed to be 0.435 from our study. This ratio is very close to 0.48 as was shown by McComiskey et
al. (2009), who also used ground-based measurements to study the marine stratus clouds over the
California coast. The logarithmic ratio (0.435) is also close to the result (0.458) of Lu et al. (2007) who
used aircraft in-situ measured cloud droplet and accumulation mode aerosol number concentration for
the marine stratus and stratocumulus clouds over the eastern Pacific Ocean. The ratio reflects the relative
conversion efficiency of cloud droplets from the CCN, regardless of the water vapor availability.
Theoretically, it has the boundaries of 0 - 1, where the lower bound means no change of $N_c$ with $N_{CCN}$,
and the upper bound indicates a linear relationship that every CCN would result in one cloud droplet.
Our result is comparable with the previous studies targeting the MBL stratiform clouds, indicating a
certain similarity of the bulk cloud microphysical responses with respect to aerosol intrusion in those
types of cloud and over different marine environments, further support that the assessment in this study
is valid.

The PWV$_{BL}$ values are represented as blue circles (larger one for higher PWV$_{BL}$) in Fig. 2a in order

to study the role of water vapor availability on the CCN-$N_c$ conversion process. As demonstrated in Fig.
2a, the PWV$_{BL}$ values almost mimic the increasing $N_{CCN,0.2\%}$ trend, which is also governed by the
seasonal $N_{CCN,0.2\%}$ and the selected cloud cases. Fig. S3 shows the seasonal variation of $N_{CCN,0.2\%}$ from
2016 to 2018. It is noticeable that the monthly $N_{CCN,0.2\%}$ values, which mimic the monthly variation of
PWV$_{BL}$, are much higher during warm months (May-Oct) than during cold months (Nov-Apr). This
seasonal $N_{CCN,0.2\%}$ variation is also found in recent studies of MBL aerosol composition and number
concentration. During the warm months, the below-cloud boundary layer is enriched by the accumulation
mode of sulfate and organic particles via local generation and long-range transport induced by the semi-
permanent Azores High, which are found to be hydrophilic and can be great CCN contributors (Wang et
al., 2020; Zawadowicz et al., 2020; Zheng et al., 2018, 2020). Therefore, the coincidence of high
$N_{CCN,0.2\%}$ and PWV$_{BL}$ does not necessarily imply a physical relationship, but instead is the result of their
similar seasonal trend. The potential co-variabilities between $N_{CCN,0.2\%}$ and PWV$_{BL}$, and hence the
implication on the $N_c$ variation will be further investigated in the latter section. When taking the PWV$_{BL}$
into account, $R^2$ increases from 0.903 to 0.982, and this new relationship suggests that the co-variability
between the binned $ln(N_{CCN,0.2\%})$ and $ln$ (PWV$_{BL}$) are in a stronger correlation with the change in
binned $ln(N_c)$. Intuitively, if the CCN-$N_c$ relationship is primarily dominated by the diffusion of water
vapor, more CCN and higher PWV$_{BL}$ should result in a continuously increasing of $N_c$. However, the
rapid increase of $N_c$ (37 to 92 cm$^{-3}$) in the first half of $N_{CCN,0.2\%}$ bins (<250 cm$^{-3}$) does not happen in
the second half of the $N_{CCN,0.2\%}$ bins (>250 cm$^{-3}$) where the slope of $N_c$ increase (96 to 103 cm$^{-3}$)
appears to be flattened for higher $N_{CCN,0.2\%}$ and PWV$_{BL}$ bins. Furthermore, the joint power-law fitting of
$N_c$ (to $N_{CCN,0.2\%}$ and PWV$_{BL}$) appears to be constantly lower than the single power-law fitting of $N_c$ (to
$N_{CCN,0.2\%}$ solely) in each bin. The negative power of PWV$_{BL}$ in this relationship suggests that PWV$_{BL}$
might play a stabilization role in the diffusional growth process, which will be further analyzed in the
following sections.
The relationship between $r_e$ and $N_{CCN,0.2\%}$ is shown in Fig. 2b where there is no significant
relationship between $r_e$ with $N_{CCN,0.2\%}$ solely, given a near-zero slope and the low correlation coefficient
(fitted line not plotted). However, after applying a multiple linear regression to the logarithmic form of
$r_e$, $N_{CCN,0.2\%}$ and $PWV_{BL}$, a significant correlation among those three variables is found. The $r_e$ is
negatively correlated with $N_{CCN,0.2\%}$ and positively correlated with $PWV_{BL}$, and 73.7% of the variations
in binned $ln\,(r_e)$ can be explained by the joint changes of the binned $ln(N_{CCN,0.2\%})$ and $ln\,(PWV_{BL})$.
This indicates that in the bulk part, $r_e$ decreases with increasing $N_{CCN,0.2\%}$ and enlarges with increasing
$PWV_{BL}$. Notice that in the lower $N_{CCN,0.2\%}$ bins ($<150$ cm$^{-3}$) where the $PWV_{BL}$ values are the lowest
among all the bins ($0.76 - 0.85$ cm), the limitation of cloud droplet growth by competing for the available
water vapor is evident by the changes in $N_c$ and $r_e$. For example, the $N_{CCN,0.2\%}$ changes from 47 to 128
cm$^{-3}$, the $N_c$ increases from 37 to 71 cm$^{-3}$ and $r_e$ only increases from 9.30 to 9.74 μm. In other words,
nearly tripling the CCN loading leads to roughly doubling $N_c$, while the $r_e$ is only enlarged by 0.44 μm
(4.7%). In the relatively low available $PWV_{BL}$ regime, it is clear that even with more CCN being
converted into cloud droplets, the limited water vapor condition prohibits the further diffusional growth
of those cloud droplets. However, in the higher $N_{CCN,0.2\%}$ bins ($>150$ cm$^{-3}$) with relatively higher
$PWV_{BL}$, the binned $r_e$ values fluctuate and decrease with increasing CCN bins under similar $PWV_{BL}$ (i.e.,
the two $N_{CCN,0.2\%}$ ranges from 200-400 cm$^{-3}$, and from 400-500 cm$^{-3}$). Since $r_e$ essentially represents
the area-weighted information of the cloud droplet size distribution (DSD), this sorting method of $r_e$
inevitably entangles multiple cloud droplet evolution processes and environmental effects that can alter
the DSD, especially under the condition of sufficient water supply. Therefore, the further assessment of
the $r_e$ responses to the $N_{CCN,0.2\%}$ loading under the constraint of water vapor should be discussed in order
to untangle the impacts of different processes and environmental effects on $r_e$.

**3.3 Aerosol-cloud interaction under different water vapor availabilities**
As previously discussed above and suggested by earlier studies, the conditions of water vapor
supply have a substantial impact on various processes from CCN-$N_c$ conversion to in-cloud droplet
condensational growth and coalescence processes, hence effectively altering the cloud DSD (Feingold et
al., 2006; McComiskey et al., 2009; Zheng et al., 2020). Moving forward to examine how $r_e$ responds to
the changes of $N_{CCN,0.2\%}$ in the context of given water vapor availability, an index describing the aerosol-
cloud interaction process is introduced as follows:
$$ACI_r = -\left.\frac{\partial \ln{(r_e)}}{\partial \ln{(N_{CCN,0.2\%})}}\right|_{PWV_{BL}} .$$    (3)
The $ACI_r$ represents the relative change of $r_e$ with respect to the relative change of $N_{CCN,0.2\%}$, where
positive $ACI_r$ denotes the decrease of $r_e$ with increasing $N_{CCN,0.2\%}$ under binned $PWV_{BL}$. This
assessment of $ACI_r$ focuses on the relative sensitivity of the cloud microphysics response in the water
vapor stratified environment, while previous studies used the cloud liquid water path (LWP) as the
constraint (Twomey, 1977; Feingold et al., 2003; Garrett et al., 2004). LWP describes the liquid water
(i.e., existing cloud droplets) physically linked to $r_e$ and $N_c$ which have an interdependent relationship
in cloud retrieval procedures, and hence to a certain extent, share co-variabilities with cloud
microphysical properties (Dong et al., 1998; Wu et al., 2020a). In this study, by using the PWV as a
sorting variable, we are trying to capture the role of ambient available water vapor in the cloud droplet
growth process (especially the water vapor diffusional growth), using measurement independent to the
cloud retrievals. Fig. 3 shows the variation of $ACI_r$ under different $PWV_{BL}$ bins, and illustrates the
calculation of $ACI_r$ in three different $PWV_{BL}$ ranges. Note that in Fig. 3a, the regressions are derived
from all points (statistically significant with a confidence level of 95%). As shown in Fig. 3a, the $ACI_r$
values range from close-to-zero values (-0.01) to 0.22, with the mean value of $0.117 \pm 0.052$. The $ACI_r$
range of this study agrees well with the previous studies of MBL cloud aerosol-cloud interactions
(McComiskey et al., 2009; Pandithurai et al., 2009; Liu et al., 2016). It is noteworthy that the variation
of $ACI_r$ with $PWV_{BL}$ suggests two different relationships under separated $PWV_{BL}$ conditions, as
discussed in the following two paragraphs.
Under the relatively lower $PWV_{BL}$ condition (<1.2 cm), the low values of $ACI_r$ (-0.01 - 0.057)
indicate that $r_e$ is less sensitive to $N_{CCN,0.2\%}$, and the dependence on $PWV_{BL}$ is also insignificant as given
by flat regression line (green dashed line) and low correlation coefficient of 0.38 (Fig. 3a). As discussed
in section 3.2, the limited water vapor can weaken the ability of condensational growth of the cloud
droplet converted from CCN, that is, the increase of CCN loading cannot be effectively reflected by a
decrease in $r_e$. For example, a 307% increase of $N_{CCN,0.2\%}$ only leads to a 10% decrease in $r_e$ in the
$PWV_{BL}$ range of 0.8-1.0 cm as shown in Fig. 3b. So that in this regime, even with a slight $PWV_{BL}$ increase,
the lack of a sufficient amount of large cloud droplets is favorable to the predominant condensational
growth process, which effectively narrows the cloud DSD and, in turn, confines the variable range of $r_e$
with respect to $N_{CCN,0.2\%}$ (Pawlowska et al., 2006; Zheng et al., 2020). In this situation, the ability of
CCN to convert to cloud droplets as well as droplet condensational growth are limited by insufficient
water vapor, rather than an influx of CCN.
However, under the relatively higher $PWV_{BL}$ regime (>1.2 cm), the $ACI_r$ values become more
positive and express a significant increasing trend with $PWV_{BL}$ (correlation coefficient of 0.83, blue
dashed line), which indicates that $r_e$ is more susceptible to $N_{CCN,0.2\%}$ in this regime. On the one hand,
due to the sufficient water vapor supply, the enhanced condensational growth process allows more CCN
to grow into cloud droplets, so that the limiting factor of the droplet growth corresponds to the changes
in CCN loading. On the other hand, the increased $N_c$ values associated with higher water vapor supply
in the cloud effectively enhance the coalescence process. This results in broadening the cloud DSD and
increasing the variation range of $r_e$ in response to the changes of $N_{CCN,0.2\%}$. To test our hypothesis of
active coalescence under higher water vapor conditions, Table 2 lists the occurrence frequencies of large
$r_e$ values (> 12 and 14 µm) under the six high $PWV_{BL}$ bins (1.2 – 2.4 cm), because this range of 12-14
µm can serve as the critical demarcation of an efficient coalescence process (Gerber, 1996; Freud and
Rosenfeld, 2012; Rosenfeld et al., 2012). As listed in Table 2, for the six high $PWV_{BL}$ bins, the
occurrence frequencies of $r_e$>12 µm are 25.0%, 30.6%, 54.1%, 74.2%, 93.8%, and 97.5%, and the
occurrence frequencies of $r_e$>14 µm are 1.25%, 1.77%, 7.4%, 17.7%, 31.9%, and 20.1%, respectively.

The increasing trends of large $r_e$ occurrences mimic the trend of $ACI_r$ and suggest that with

increased $PWV_{BL}$, cloud droplets have a greater chance to grow via the effective coalescence process
and subsequently lead to an enlargement of $ACI_r$. Although previous studies have brought up the
potential impacts of the cloud droplet coalescence process on ACI, it is rarely seen that the relationship
among them has been discussed in detail. Here we provide possible explanations on how the enhanced
coalescence process can enlarge $ACI_r$. Quantitatively, $ACI_r$ is described by the logarithmic partial
derivative ratio of $r_e$ to $N_{CCN,0.2\%}$, thus a sharper decrease of $r_e$ with respect to a given $N_{CCN,0.2\%}$ range
can result in a steeper slope and in turn, larger $ACI_r$ (i.e., a 239% increase in $N_{CCN,0.2\%}$ leads to a $r_e$
decrease of 48% in the 2.2-2.4 cm bin in Fig. 3b). Physically, this relies on how the cloud droplet size
distribution (DSD) would change with different CCN loadings. Therefore, particularly in low CCN
conditions, sufficient water vapor availability will allow cloud droplets to continuously grow via
diffusion of water vapor (i.e., condensational growth), and enter the active cloud-droplet coalescence
regime. In contrast, the increase in cloud droplet size can effectively reduce $N_c$ via the process of large
cloud droplets collecting small droplets, and small droplets be coalesced into large droplets.
Consequently, the cloud DSD becomes effectively broadened toward the large tail by the coalescence,
so that $r_e$ is enlarged. With more CCN available, the cloud DSD is narrowed by the enhanced
condensational growth and regresses toward the small tail by increasing the amount of newly converted
cloud droplets which result in decreased $r_e$. These interactions between CCNs and cloud droplets
ultimately result in the broadened changeable range of $r_e$, and in turn, the enlarged $ACI_r$.

413 In order to investigate the theoretical implication of supersaturation conditions on the aerosol-cloud
414 interaction observed here in the MBL stratiform clouds, the $ACI_r$ values are calculated with respect to
415 the surface $N_{CCN}$ theoretically at two additional high supersaturation levels (0.5% and 1.2%), under all
416 $PWV_{BL}$ conditions. The results in Table 3 show that the $ACI_r$ signals are both weak and do not have
417 significant changes under relatively lower $PWV_{BL}$ conditions, while the $ACI_r$ signals tend to strengthen
418 with the increase of supersaturation under the relatively higher $PWV_{BL}$. Based on Köhler theory, if the
419 supersaturation exceeds the critical point for the given droplet, the droplet will thus experience continued
420 growth, so theoretically the ACI should increase with the supersaturation under same aerosol number
421 concentration. However, the observed limited water vapor cannot support this ideal droplet growth,
422 results in weak responses of cloud droplets to aerosol intrusion. With the increase of observed water
423 vapor, the continued growth of cloud droplets becomes more plausible, hence the high supersaturation
424 yields larger droplets with low number of aerosols, more efficient droplet activation with a large number
425 of aerosols, and in turns, larger $ACI_r$ (even out of the theoretical bounds). However, considering these
426 high supersaturation environments are unphysical in the observed MBL cloud layers, and estimating the
427 real supersaturation conditions using ground-based remote-sensing is beyond the scope of this study, we
428 chose the supersaturation level of 0.2% because it represents the most typical supersaturation conditions
429 of MBL stratiform clouds.

431 **3.4 The co-variabilities of the meteorological factors**

432 The environmental conditions over the ENA have been widely studied as not independent but
433 entangled with each other (Wood et al., 2015; Zheng et al., 2016; Wu et al., 2017; Wang et al., 2021).
434 To better understand the dependencies and the co-variabilities of the meteorological factors, a principal
435 component analysis (PCA) is performed comprising the following variables: (1) $PWV_{BL}$ denotes the
436 water vapor availability within the boundary layer; (2) $D_i$ describes the boundary layer coupling
437 conditions; (3) $TKE_w$ represents the strength of boundary layer turbulence; (4) $W_{dir,NS}$ reflects the
438 surface wind directions in terms of northerly and southerly; and (5) LTS infers the large-scale
439 thermodynamic structures. Note that the $W_{dir,NS}$ are taken as $W_{dir,NS} = abs(W_{dir} - 180°)$, so that the
440 original $W_{dir}$ (0-360°) can be transformed to $W_{dir,NS}$ (0-180°) where the values smaller than 90° are
441 close to the southerly wind, and those greater than 90° are close to the northerly wind. The $W_{dir,ns}$ are
442 transformed as such to capture the island effects better, because the cliff is located north of the ENA site.
443 The input data metric of the PCA is constructed from the above five variables, thus the principal
444 components (PCs) that explaining the variations of those dependent variables can be output from the

eigenanalysis. The result shows that for the five selected meteorological factors, the proportions of the
total intervariable variance explained by the PCs are 43.72%, 22.01%, 18.26%, 8.95% and 7.06%, and
the eigenvalues are 2.19, 1.10, 0.91, 0.45, and 0.35, respectively. Note that the first three PCs have the
highest eigenvalues and explain most (~84%) of the total variance, which indicates that they can capture
the significant variation patterns of the selective meteorological factors.

To determine the relative contributions of the variables to PCs, all the five selected meteorological

variables are projected to the first three PCs and the Pearson correlation coefficients between them are
listed in Table 4. For the first PC (PC1) which accounts for the highest proportion (43.72%) of the total
variance, the PC1 is strongly negatively correlated with $PWV_{BL}$ (-0.84) and $D_i$ (-0.73), but strongly
positively correlated with $TKE_w$ (0.69). These results suggest that PC1 mainly represents the boundary
layer conditions, and the co-variations of the boundary layer water vapor and turbulence are the most
distinct environmental patterns for the selected cloud cases. The PC2 and PC3 are most correlated with
LTS (0.58 and 0.65 for PC2 and PC3, respectively) and $W_{dir,NS}$ (0.60 and -0.50 for PC2 and PC3,
respectively), indicating that the PC2 and PC3 mainly describe the variations in large-scale
thermodynamic and the surface wind patterns, which are likely associated with the variations of the
Azores High position and strength (Wood et al., 2015).

To further understand the correlations between the meteorological variables, the principal

component loadings plot is constructed by projecting the variables onto PC1 and PC2 as shown in Fig.
4. Each point denotes the variable correlations with PC1 (x-coordinate) and PC2 (y-coordinate), so that
each vector represents the strength and direction of the original variable influences on the pair of PCs.
The angle between the two vectors represents the correlation between each other. In Fig. 4, both $TKE_w$
and $W_{dir,NS}$ vectors are located in the same quadrant (positive in both PC1 and PC2) and close to each
other with a small degree of an acute angle, which means the $TKE_w$ are strongly correlated with the
$W_{dir,NS}$. When the surface wind is coming from the north side of the island, the topographic lifting effect
of the cliff would induce additional updraft over the ENA site (Zheng et al., 2016), so that the wind closer
to the northerly wind (larger $W_{dir,NS}$) is more correlated with higher $TKE_w$. Note that $TKE_w$ and $D_i$
vectors are almost in an opposite direction, which denotes a strongly negative correlation between the
two variables. The angles of $PWV_{BL}$ with $D_i$ (~45°) and $TKE_w$ (~142°) suggest that $PWV_{BL}$ is
moderately positively correlated with $D_i$ but negatively correlated with $TKE_w$. A higher $D_i$ indicates a
more decoupled MBL, where MBL is not well-mixed and separated into a radiative-driven layer and a
surface flux driven layer that caps the surface moisture (Jones et al., 2011). This situation is more likely
to be associated with a relatively higher $PWV_{BL}$ and weaker $TKE_w$ condition. Note that the negative
correlation between $D_i$ and $TKE_w$ examined here might also be partly attributed to the diurnal cycle of
the turbulence, which is studied to be associated with the cloud-top longwave radiative cooling over the
ENA, especially for the drizzling clouds (Ghate et al., 2021; Zheng et al., 2016). However, this study
focuses on the non-precipitating clouds where the effect of drizzle on the cloud-top radiative cooling
driven turbulence is minimum, and examining the cloud-top radiative cooling rate from ground-based
remote sensing is beyond the scope of the current study. It would be with interest to get the accurate
cloud-top radiative cooling rate using a radiative transfer model to perform further study in the future.
As for the LTS parameter, the close to 90° angle with $TKE_w$ suggests no correlation between them, since
the LTS is mostly capturing the large-scale thermodynamical structures and is obtained from a coarser
temporal resolution. Thus, the LTS does not essentially have correspondence to the strength of boundary
layer turbulence and can be treated as independent to $TKE_w$ over the ENA site. The loading plot
intuitively tells us the directions and strengths of the co-variabilities of the selected meteorological
variables, and sheds the light on determining the key factors that are feasible to use in examining the
environmental impacts on the aerosol-cloud interactions.
**3.5 Linking the meteorological factors to aerosol-cloud interaction**
**3.5.1 Relations of meteorological factors with aerosol and cloud properties**
The PCs are, mathematically, the linear combination of the selected variables, and hence independent of
each other after the PCA. Therefore, treating the aerosol and cloud properties as dependents and
correlated with the PCs allows us to infer their co-variation with the meteorological factors statistically.
A weakly negative correlation between $N_{CCN,0.2\%}$ and PC1 ($R_{PC1,CCN} = -0.35$) suggests that the
relatively higher $N_{CCN,0.2\%}$ could be sometimes found under higher $PWV_{BL}$ and lower $TKE_w$. Though
the correlation is low, the plausible contributions could come from the seasonal variations of $N_{CCN,0.2\%}$
and $PWV_{BL}$ as discussed in the previous section, and the weaker $TKE_w$ might prevent the vertical mixing
of CCN and induce higher surface $N_{CCN,0.2\%}$. On the other hand, a weakly positive correlation between
$N_{CCN,0.2\%}$ and PC2 ($R_{PC2,CCN} = 0.21$) suggests that there are no fundamental relationships between
CCN with thermodynamic and the surface wind direction, and they are not the key controlling factor of
surface $N_{CCN,0.2\%}$ variation because the surface CCN concentration is primarily contributed by the
accumulation-mode aerosols which come from the condensational growth of Aitken-mode aerosols
(Zheng et al., 2018). As for the cloud properties, both $N_c$ and $f_{ad}$ are negatively correlated with PC1
($R_{PC1,Nc} = -0.51$ and $R_{PC1,fad} = -0.62$, respectively), suggesting a moderate relationship between $N_c$,
$f_{ad}$, and the boundary layer condition. These negative correlations suggest that under the higher $PWV_{BL}$
condition, the sufficient water vapor supply allows more CCN to become cloud droplets, as previously
discussed, and hence increases the cloud adiabaticity due to the dominant condensational growth process.
While in the situation of relatively higher $TKE_w$, the decrease in the $N_c$ and $f_{ad}$ might be partly attributed
to the association with the active in-cloud coalescence process and entrainment of dry air. However,
owing to the obstacle of retrieving in-cloud $TKE_w$ from the ground-based remote sensing, the usage of
sub-cloud $TKE_w$ in this study captures part of the relationship between turbulence and adiabaticity.
Therefore, in this situation, the cloud adiabaticity might depend more on $PWV_{BL}$ and the boundary layer
decoupling state. Moreover, their low correlations with PC2 ($R_{PC2,Nc} = -0.10$ and $R_{PC2,fad} = -0.17$,
respectively) indicate very weak relations with the large-scale thermodynamic variables. These weak
correlations might likely be due to the subset of MBL single-layer stratocumulus in this study, as the
previous study over the ENA found that the sensitivity of MBL cloud adiabaticity largely depends on the
strength of cloud top inversion (which can be partially indicated by the increased LTS) and slightly
depends on the boundary layer decoupling (Terai et al., 2019; Zheng et al., 2020). Note that the same
sign of correlations with PC1 statistically infer the similar directional co-variation of $N_{CCN,0.2\%}$, $N_c$, and
$f_{ad}$ to a certain extent.

To examine the physical relation between $N_{CCN,0.2\%}$, $N_c$ and $f_{ad}$, the profiles of cloud $r_e$ and

LWC are plotted in normalized height from cloud base ($z_b$) to cloud top height ($z_t$) (Fig. 5), which is
given by $z_n = (z - z_b) / (z_t - z_b)$. The solid lines denote the mean values, and the shaded area
represents one standard deviation at each normalized height $z_n$. The normalized $r_e$ increases from ~8.6
$\mu m$ at the cloud base toward ~11 $\mu m$ near the upper part of the cloud where $z_n$ is 0.7 (Fig. 5a), through
condensational growth and coalescence processes, and then decreases toward the cloud top due to cloud-
top entrainment. Similar in-cloud vertical variation of $r_e$ is also found by previous study using aircraft
in-situ measurements (Zhao et al., 2018; Wu et al. 2020a). Profiles of retrieved LWC and calculated
adiabatic $LWC_{ad}$ (blue line) are presented in Fig. 5b. As demonstrated in Fig. 5b, the $f_{ad}$ values, which
is the ratio of LWC to $LWC_{ad}$, reach a maximum of 0.8 at the cloud base and a minimum of 0.38 at the
cloud top. The shaded areas of $r_e$ and LWC denote the range from near-adiabatic to sub-adiabatic cloud
environments, where in the near-adiabatic cloud (higher $f_{ad}$) the cloud droplets experience adiabatic
growth and LWC should be close to $LWC_{ad}$. In contrast, in the sub-adiabatic cloud regime, the decrease
of $f_{ad}$ is largely due to cloud-top entrainment and coalescence processes even in non-precipitating MBL
clouds (Wood, 2012; Braun et al., 2018; Wu et al. 2020b). Furthermore, to understand the implication of
cloud adiabaticity with respect to CCN-$N_c$ conversion, all of the $f_{ad}$ samples are separated into two
groups by the median value of the layer-mean $f_{ad}$ (0.66) for further analysis.
Figure 6 shows $N_c$ against the binned $N_{CCN,0.2\%}$ for the near-adiabatic regime ($f_{ad} > 0.66$) and
sub-adiabatic regime ($f_{ad} < 0.66$). For the near-adiabatic regime, $N_c$ increases from ~60 cm$^{-3}$ to 119
cm$^{-3}$ with increased $N_{CCN,0.2\%}$ and PWV$_{BL}$, and both $N_{CCN,0.2\%}$ and PWV$_{BL}$ appear to play positive roles
in terms of the $N_c$ increase. The result is as expected because the process of condensational growth is
predominant in the near-adiabatic clouds, that is, with increasing water vapor supply, the higher CCN
loading can effectively lead to more cloud droplets. However, in the sub-adiabatic cloud regime, $N_c$
increases with increased $N_{CCN,0.2\%}$ but possesses a negative correlation with PWV, which results in a
slower increase of $N_c$ under higher $N_{CCN,0.2\%}$ and PWV$_{BL}$ conditions. The mean reduction of $N_c$ in the
sub-adiabatic regime is computed to be ~37% compared to that for the near-adiabatic clouds. As
previously studied, the coalescence process contributes significantly to $N_c$ depletion, even in a non-
precipitating MBL clouds (Feingold et al., 1996; Wood, 2006). Thus, lower $N_c$ in the sub-adiabatic
regime may be partly due to the combined effect of coalescence and entrainment (Wood, 2006; Hill et
al., 2009; Yum et al., 2015; Wang et al., 2020). Note that the retrieved $N_c$ represents the cloud layer-
mean information. In summary, the Wu et al. (2020a) retrieval works to separate the reflectivity into the
contributions of cloud ($Z_c$) and drizzle. The retrieval assumes an initial guess of the representative layer-
mean $N_c$ based on the climatology over ENA sites (Dong et al., 2014), and such allows the first guess of
the vertical profile of LWC based on $N_c$ and $Z_c$, and then constrains the $N_c$ and LWC using the LWP
derived from MWR, and finally output $r_e$ values (Fig. 3 in Wu et al., 2020a). Therefore, the final
retrieved $N_c$ is updated to in response to the cloud microphysical processes within this time-step. From
the aircraft in-situ measurements during the ACE-ENA, we found that the observed vertical profile of
$N_c$ is near-constant in the middle part of the cloud (even in the drizzling cloud where the collision-
coalescence processes are more active), and the signal of entrainment-induced $N_c$ depletion is shown
near the cloud top (Wu et al., 2020a). However, it is difficult and beyond the scope of the ground-based
retrieval to compare the vertical dependency of depletion rate within one time-step. Therefore, as the
retrieval currently works to represent the layer-mean information from the given time-step, the preferred
method in this study is to compare $N_c$ at different times, where in this case are the adiabatic versus sub-
adiabatic conditions which hence yields different $N_c$ that we retrieved from the ground-based snapshot
perspective. From the PCA and binning analysis, the effect of cloud adiabaticities on CCN- $N_c$
conversions may shed light on interpreting the aerosol-cloud interaction under different environmental
effects.

**3.5.2 The role of meteorological factors on ACI$_r$ assessment**

Since ACI$_r$ can only be calculated by the logarithmic derivatives from a set of $N_{CCN,0.2\%}$ and $r_e$ data within a certain regime, it will be inappropriate to linearly correlate the data with PCs directly, in both mathematical and physical perspectives. Therefore, the meteorological factors which have the strongest influence on the most explanatory PCs, namely PWV$_{BL}$ and TKE$_w$ are selected to be the sorting variables in assessing the environmental impacts on the ACI$_r$. In addition, LTS is also selected as it represents the large-scale thermodynamic factor and is independent to the boundary-layer environment conditions. The data samples are first separated into two regimes using the median values of the targeting factors, and then separated into four quadrants by the median PWV$_{BL}$ because ACI$_r$ is found to have significant differences under different water vapor availabilities. The ACI$_r$ values are further calculated for all quadrants to examine whether the ACI$_r$ can be distinguished by the targeting factors.

Combining LTS and PWV$_{BL}$ as sorting variables, the ACI$_r$ values for four regimes are shown in Fig. S4. The ACI$_r$ differences between low and high PWV$_{BL}$ regimes are still retained. In the low PWV$_{BL}$ regime, the ACI$_r$ values are limited to 0.016 and 0.056 for low and high LTS regimes, respectively. In the high PWV$_{BL}$ regime, the ACI$_r$ values are 0.150 and 0.171 for low and high LTS regimes, respectively, which is about 3-5 times greater than those in low PWV$_{BL}$ regime. However, the ACI$_r$ in different LTS regimes cannot be distinctly differentiated (ACI$_r$ differences between LTS regimes are ~0.02 and ~0.04), and the main difference in ACI$_r$ are still induced by the PWV$_{BL}$. Owing to the location of the ENA site where it locates near the boundary of mid-latitude and subtropical climate regimes, the MBL clouds over the ENA are found to be often under the influences of cold fronts associated with mid-latitude cyclones, where the cloud evolutions are subject to the combine effects of post-frontal and large-scale subsidence (Wood et al., 2015; Zheng et al., 2020; Wang et al., 2021). Therefore, over the ENA, although the spatial gradient of LTS is studied to be associated with the production of MBL turbulence and the change in wind direction (Wu et al., 2017), the LTS value itself is examined to have a weak impact on the aerosol-cloud interaction from this study.

The TKE$_w$ has been found to be strongly positively correlated with $W_{dir,NS}$ and negatively correlated with $D_i$ from the PCA, that is, the values of TKE$_w$ already account for the co-variabilities in these variables. Therefore, treating TKE$_w$ as the sorting variable would lead to a more physical process-orientated assessment. Accordingly, to examine the role of the dynamical factors on ACI, the samples are separated into four regimes demarcated by the median values of PWV$_{BL}$ and TKE$_w$ (Fig. 7), and the mean values of $D_i$ and $f_{ad}$ in the four quadrants are also displayed in Fig. 7. The effect of PWV$_{BL}$ on

$ACI_r$ is demonstrated by the mean $ACI_r$ values where they are much higher in the high $PWV_{BL}$ regime
than those in the low $PWV_{BL}$ regime no matter what the $TKE_w$ regimes. Furthermore, the result illustrates
that $TKE_w$ does play an important role in $ACI_r$, because the $ACI_r$ values in the high $TKE_w$ regime are
more than double than the values in the low $TKE_w$ regime.

In the regimes of high $TKE_w$ and $PWV_{BL}$, which are closely associated with coupled MBL ($D_i =$

$0.21$) and more sub-adiabatic cloud conditions ($f_{ad} = 0.52$), $r_e$ is highly sensitive to CCN loading with
the highest $ACI_r$ of 0.259. The sufficient water vapor availability allows CCN to be converted into cloud
droplets more effectively, while the relatively higher $TKE_w$ indicates stronger turbulence in the below-
cloud boundary layer and maintains a nearly well-mixed MBL. The CCN and moisture below-cloud layer
are efficiently transported and mixed aloft via the ascending branch of the eddies (Nicholls, 1984; Hogan
et al., 2009), hence are effectively connected to the cloud layer. Therefore, under the lower CCN loading
condition, the active coalescence process (which indicated by the low $f_{ad}$ values) results in the depletion
of small cloud droplets and broadening of cloud DSD (Chandrakar et al., 2016), and in turn, leads to
further enlarged $r_e$. However, with higher CCN intrusion into the cloud layer, the enhanced cloud droplet
conversion and the subsequential condensational growth behave contradictorily to narrow the DSD
(Pinsky and Khain, 2002; Pawlowska et al., 2006), which leads to decreased $r_e$. Therefore, the MBL
clouds are distinctly susceptible to CCN loading under the environments of sufficient water vapor and
strong turbulence in which the $ACI_r$ is enlarged.

Under high $PWV_{BL}$ but low $TKE_w$ conditions, the mean $ACI_r$ reduces to 0.101 (~ 39% of that

under high $TKE_w$). The MBL is more likely decoupled where $D_i = 0.54$, which indicates that the weaker
turbulence loosens the connection between the cloud layer and the underlying boundary layer. This
results in a less effective conversion of CCN into cloud droplets, while the more adiabatic cloud
environment ($f_{ad} = 0.75$) denotes the lack of coalescence growths and thus diminishes the $r_e$ sensitivity
to CCN. Although the constraints of insufficient water vapor on $ACI_r$ are still evident, the $ACI_r$ values
increase from 0.008 in the low $TKE_w$ regime to 0.024 in the high $TKE_w$ regime. The $ACI_r$ differences
between the two $TKE_w$ regimes attest that $ACI_r$ strongly depends on the connection between the cloud
layer and the below-cloud boundary layer CCN and moisture, that is, stronger turbulence can enhance
the susceptibility of $r_e$ to CCN.

In this study, the relationship between turbulence and ACI is found to be valid in non-precipitating

MBL clouds. Theoretically, the effect of turbulence on $ACI_r$ would appear to be artificially amplified, if
in the presence of precipitation. The intensive turbulence can enhance the coalescence process and
accelerate the CCN-cloud cycling, and subsequently, the CCN depletion due to precipitation and
coalescence scavenging would result in quantitatively enlarged $ACI_r$ (Feingold et al., 1996, 1999; Duong
et al., 2011; Braun et al., 2018). Though it is beyond the scope of this study, it would be of interest to
perform such analysis on the aerosol-cloud-precipitation interaction using ground-based remote sensing
and model simulations in a future study.

**4.  Summaries and Conclusions**

Over the ARM-ENA site, a total of 20 non-precipitating single-layered MBL stratus and
stratocumulus cloud cases have been selected in order to investigate the aerosol-cloud interaction (ACI).
The distributions of CCN and cloud properties for selected cases represent the typical characteristics of
non-precipitating MBL clouds in a relatively clean environment over the remote oceanic area. The
diversity of boundary layer conditions and cloud adiabaticities among the selected cases enable the
investigation of different environmental effects on ACI.

The overall variations of $N_c$ with $N_{CCN,0.2\%}$ show an increasing trend, regardless of the water vapor
condition, while the sufficient $PWV_{BL}$ appears to stabilize the CCN-$N_c$ conversion process. The water
vapor limitation on cloud droplet growth is evident in the lower $N_{CCN,0.2\%}$ up to 150 cm$^{-3}$ with low
$PWV_{BL}$ values, where a near tripling of CCN loading leads to a near doubling of $N_c$ but only 4.7%
increase in $r_e$. When $N_{CCN,0.2\%}$ is greater than 250 cm$^{-3}$ and $PWV_{BL}$ values are also relatively high, $r_e$
appears to decrease with increasing $N_{CCN,0.2\%}$ under similar water vapor conditions. As for bulk aerosol-
cloud interaction, the $ACI_r$ values vary from -0.01 to 0.22 for different $PWV_{BL}$ conditions where $ACI_r$
appears to be diminished under limited water vapor availability due to limited droplet activation and
condensational growth processes. While under relatively sufficient water supply conditions, $r_e$ shows
more sensitive responses to the changes of $N_{CCN,0.2\%}$, due to the combined effect of condensational
growth and coalescence processes accompanying the higher $N_c$ and $PWV_{BL}$.

The theoretical diagram describing the mechanism proposed above is shown in Fig. 8. Under the
relatively lower $PWV_{BL}$ condition, the limited water vapor weakens the ability of condensational growth
of the cloud droplet converted from CCN, which results in both less newly converted as well as large
cloud droplets, with the lack of chance of coalescence processes under this circumstance. Therefore, the
variable range of $r_e$ versus $N_{CCN,0.2\%}$ is narrowed and presented as small $ACI_r$. While under the relatively
higher $PWV_{BL}$ condition, particularly in low CCN conditions, the sufficient water vapor availability
allows cloud droplets growing via the condensation of water vapor, and thus enter the active cloud-
droplet coalescence regime. In contrast, the increase in cloud droplet size can effectively reduce $N_c$ via
the coalescence process and the size distributions are effectively broadened toward the large tail by the
coalescence, so that $r_e$ is enlarged. Under a higher $N_{CCN,0.2\%}$ intrusion, the cloud droplet size distribution
is narrowed by the enhanced condensational growth and regresses toward the small tail by increasing the
amount of newly converted cloud droplets which results in decreased $r_e$. Combinedly, the interactions
between CCNs and cloud droplet growth processes ultimately result in a broadened changeable range of
$r_e$, and in turn, the enlarged $ACI_r$.
The co-variabilities among the environmental factors are examined using the multi-dimensional
PCA. The variables of $PWV_{BL}$, $D_i$, $TKE_w$, LTS and $W_{dir,NS}$ are constructed as the input of the
eigenanalysis. Results show that the first three PCs can describe the majority (~84%) of the variance
among the selected variables. The most explanatory PC1 (account for 43.72% contribution) strongly
correlated with $PWV_{BL}$, $D_i$ (both negatively) and $TKE_w$ (positively), and hence describe the co-variation
of the boundary layer conditions. While the PC2 and PC3 (account for 22.01% and 18.26% contributions,
respectively) are strongly correlated with LTS and $W_{dir,NS}$, which likely indicates the variations of the
Azores High position and strength. By projecting the variables onto PC1 and PC2, the PCA loading
analysis shows that $TKE_w$ is strongly negatively correlated with $D_i$, which is what we expected. A
decoupled MBL cloud is often separated into two layers where the lower one can cap the surface moisture,
while the higher $TKE_w$ denote sufficient turbulence that maintains the well-mixed MBL. Additionally,
the island effect is also indicated by the eigenanalysis, where surface northerly wind would induce
additional updraft velocity and hence disturb $TKE_w$, owing to the effect of the cliff north of the ENA site.
The role of cloud adiabaticities on the behaviors of CCN-$N_c$ conversion is examined using both binning
and eigenanalysis. In a near-adiabatic cloud vertical structure, the cloud droplet growth process is
dominated by condensational growth, thus the $N_c$ responses to increased $N_{CCN,0.2\%}$ and $PWV_{BL}$ are
strengthened. When the cloud layer becomes more sub-adiabatic, the effect of coalescence leads to the
depletion of $N_c$ and thus results in the lower retrieved $N_c$ from a ground-based snapshot perspective. The
competition between the condensational growth and coalescence processes strongly impacts the
variations of cloud microphysics to CCN loading.
To investigate the environmental effects on $ACI_r$, the factors having the most influence on the
explanatory PCs are selected as the sorting variables in the $ACI_r$ assessments. The LTS sorting method
cannot distinguish the $ACI_r$ values, which means the LTS values themselves have a weak impact on $ACI_r$
due to the MBL cloud cover over the ENA is mainly impacted by the mid-latitude cyclone systems. In
contrast, the intensity of boundary layer turbulence represented by $TKE_w$ plays a more important role in
$ACI_r$, since the values of $TKE_w$ already account for the co-variations of the MBL conditions, and hence
leads to a physical process-oriented assessment. The $ACI_r$ assessments in four different $TKE_w$ and
PWV$_{BL}$ regimes show that the constraints of insufficient water vapor on the ACI$_r$ are still evident, but in
both PWV$_{BL}$ regimes the ACI$_r$ values increase more than double from low TKE$_w$ to high TKE$_w$ regimes.
Noticeably, the ACI$_r$ increases from 0.101 in the low TKE$_w$ regime to 0.259 in the high TKE$_w$ regime,
under high PWV$_{BL}$ conditions. The intensive below-cloud boundary layer turbulence strengthens the
connection between the cloud layer and below-cloud CCN and moisture. So that with sufficient water
vapor, an active coalescence leads to further enlarged $r_e$, particularly for low CCN loading conditions,
while the enhanced $N_c$ from condensational growth induced by increased $N_{CCN,0.2\%}$ can effectively
decrease $r_e$. Combining these processes together, the enlarged ACI$_r$ is presented.
In this study, the non-precipitating MBL clouds are found to be most susceptible to the below-cloud
CCN loading under environments with sufficient water vapor and stronger turbulence. This study
examines the importance of the environmental effects on the ACI$_r$ assessments, and provides the
observational constraints to the future model evaluations on the aerosol-cloud interactions. Future studies
will be focusing on exploring the role of environmental effects on the aerosol-cloud-precipitation
interactions in MBL stratocumulus through an integrative analysis of observations and model simulations.

*Data availability.* Data used in this study can be accessed from the DOE ARM's Data Discovery at
https://adc.arm.gov/discovery/

*Author contributions.* The original idea of this study is discussed by XZ, BX, and XD. XZ performed the
analyses and wrote the manuscript. XZ, BX, XD, PW, YW and TL participated in further scientific
discussions and provided substantial comments and edits on the paper.

*Competing interests.* The authors declare that they have no conflict of interest.

*Special issue statement.* This article is part of the special issue "Marine aerosols, trace gases, and clouds
over the North Atlantic (ACP/AMT inter-journal SI)". It is not associated with a conference.

*Acknowledgments.* The ground-based measurements were obtained from the Atmospheric Radiation
Measurement (ARM) Program sponsored by the U.S. Department of Energy (DOE) Office of Energy
Research, Office of Health and Environmental Research, and Environmental Sciences Division. The
reanalysis data were obtained from the ECMWF model output, which provides explicitly for the analysis
at the ARM ENA site. The data can be downloaded from https://adc.arm.gov/discovery/. This work was
supported by the NSF grants AGS-1700728/1700727 and AGS-2031750/2031751, and was also
supported as part of the "Enabling Aerosol-cloud interactions at GLobal convection-permitting scalES
(EAGLES)" project (74358), funded by the U.S. Department of Energy, Office of Science, Office of
Biological and Environmental Research, Earth System Modeling program with the subcontract to the
University of Arizona. The Pacific Northwest National Laboratory is operated for the Department of
Energy by Battelle Memorial Institute under Contract DE-AC05-76 RL01830. And a special thanks to
coeditor Dr. Hang Su and three anonymous reviewers for the constructive comments and suggestions,
which helped to improve the manuscript.

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

**Table 1.** Dates and time periods of selected non-precipitating MBL cloud periods

| Case No. | Start Date | Start UTC | End Date | End UTC | Valid Samples |
|---|---|---|---|---|---|
| 1 | 20160915 | 2200 | 20160916 | 0020 | 24 |
| 2 | 20170219 | 2110 | 20170220 | 0520 | 87 |
| 3 | 20170222 | 0830 | 20170222 | 1200 | 38 |
| 4 | 20170605 | 1430 | 20170605 | 1900 | 54 |
| 5 | 20170616 | 1230 | 20170616 | 1510 | 32 |
| 6 | 20170617 | 0320 | 20170617 | 0520 | 24 |
| 7 | 20170627 | 0020 | 20170627 | 0250 | 28 |
| 8 | 20170630 | 0530 | 20170630 | 0930 | 42 |
| 9 | 20170630 | 1400 | 20170630 | 1700 | 34 |
| 10 | 20170706 | 0140 | 20170706 | 0900 | 62 |
| 11 | 20170707 | 0130 | 20170707 | 1000 | 91 |
| 12 | 20170910 | 2100 | 20170911 | 0600 | 94 |
| 13 | 20170911 | 1930 | 20170911 | 2150 | 24 |
| 14 | 20170912 | 0820 | 20170912 | 1100 | 32 |
| 15 | 20171006 | 2110 | 20171006 | 2320 | 26 |
| 16 | 20180130 | 1030 | 20180131 | 0500 | 152 |
| 17 | 20180203 | 1930 | 20180204 | 0500 | 72 |
| 18 | 20180324 | 0210 | 20180324 | 0600 | 46 |
| 19 | 20180508 | 0730 | 20180508 | 1110 | 42 |
| 20 | 20180513 | 2130 | 20180514 | 1200 | 139 |

**Table 2.** Occurrence frequencies of large in-cloud $r_e$ * under relatively high PWV conditions

| PWV (cm) | 1.2-1.4 | 1.4-1.6 | 1.6-1.8 | 2.8-2.0 | 2.0-2.2 | 2.2-2.4 |
|---|---|---|---|---|---|---|
| $r_e > 12$ μm (%) | 25.0 | 30.6 | 54.1 | 74.2 | 93.8 | 97.5 |
| $r_e > 14$ μm (%) | 1.25 | 1.77 | 7.4 | 17.7 | 31.9 | 20.1 |

*The occurrence of large $r_e$ is defined when the $r_e$ is found to be larger than 12 μm or 14 μm using the retrieved in-cloud vertical profiles.

**Table 3.** $ACI_r$ calculated with respect to $N_{CCN}$ theoretically at different supersaturation levels, under all $PWV_{BL}$ conditions

| $PWV_{BL}$ (cm) | 0.4-0.6 | 0.6-0.8 | 0.8-1.0 | 1.0-1.2 | 1.2-1.4 | 1.4-1.6 | 1.6-1.8 | 1.8-2.0 | 2.0-2.2 | 2.2-2.4 |
|---|---|---|---|---|---|---|---|---|---|---|
| $ACI_r$ ($N_{CCN}$@0.2%SS) | 0.020 | 0.057 | 0.002 | -0.014 | 0.108 | 0.076 | 0.145 | 0.151 | 0.221 | 0.175 |
| ($N_{CCN}$@0.5%SS) | 0.023 | 0.057 | 0.0002 | 0.024 | 0.129 | 0.121 | 0.309 | 0.136 | 0.293 | 0.159 |
| ($N_{CCN}$@1.2%SS) | 0.023 | 0.045 | 0.002 | 0.072 | 0.125 | 0.123 | 0.323 | 0.175 | 0.347 | 0.186 |

**Table 4.** The first three principal components from eigenanalysis

| Eigenanalysis | PC1 | PC2 | PC3 |
|---|---|---|---|
| Eigenvalues | 2.17 | 1.10 | 0.91 |
| Proportion of variance explained (%) | 43.72 | 22.01 | 18.26 |
| Cumulative proportion (%) | 43.72 | 65.73 | 83.99 |
| Correlations (Variables vs. PCs) | PC1 | PC2 | PC3 |
| $PWV_{BL}$ | -0.84 | 0.20 | -0.11 |
| $D_i$ | -0.73 | -0.48 | -0.20 |
| $TKE_W$ | 0.69 | 0.35 | -0.44 |
| $W_{dir,ns}$ | 0.52 | 0.60 | -0.50 |
| LTS | -0.43 | 0.58 | 0.65 |

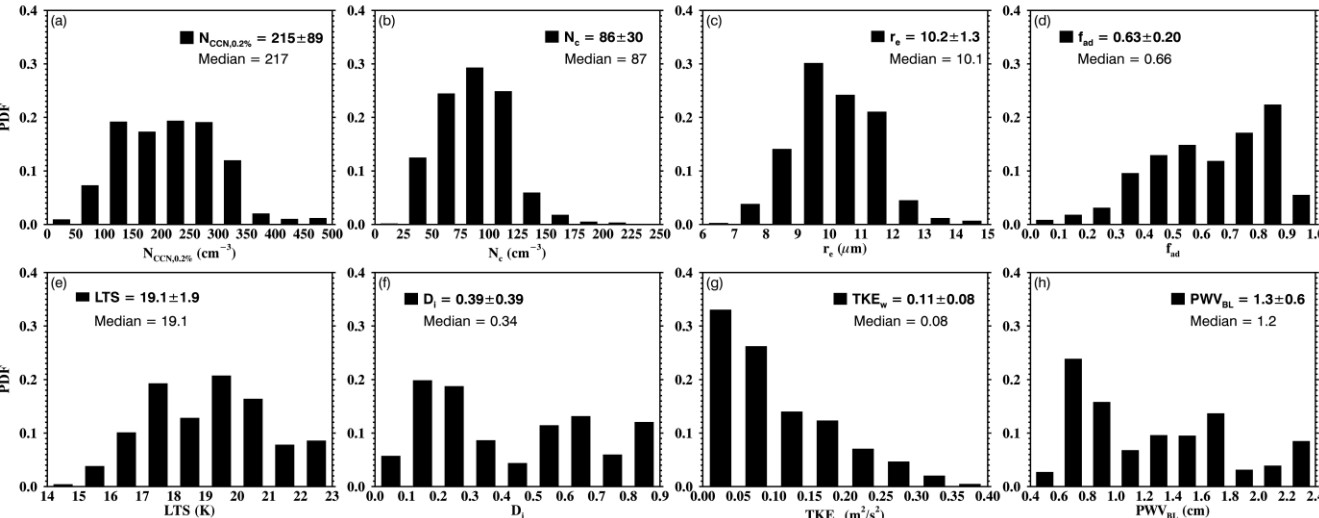

**Figure 1.** Probability distribution functions (PDFs), mean, standard deviation and median values of aerosol, cloud, and meteorological properties for 20 selected non-precipitating cloud cases at the DOE ENA site during the period 2016-2018. (a) Cloud condensation nuclei (CCN) number concentration at 0.2% supersaturation ($N_{CCN,0.2\%}$); (b) cloud-droplet number concentration ($N_c$); (c) cloud-droplet effective radius ($r_e$); (d) cloud adiabaticity ($f_{ad}$); (e) lower tropospheric stability (LTS); (f) decoupling index ($D_i$); (g) mean vertical component of turbulence kinetic energy ($TKE_w$); and (h) sub-cloud boundary-layer precipitable water vapor ($PWV_{BL}$).

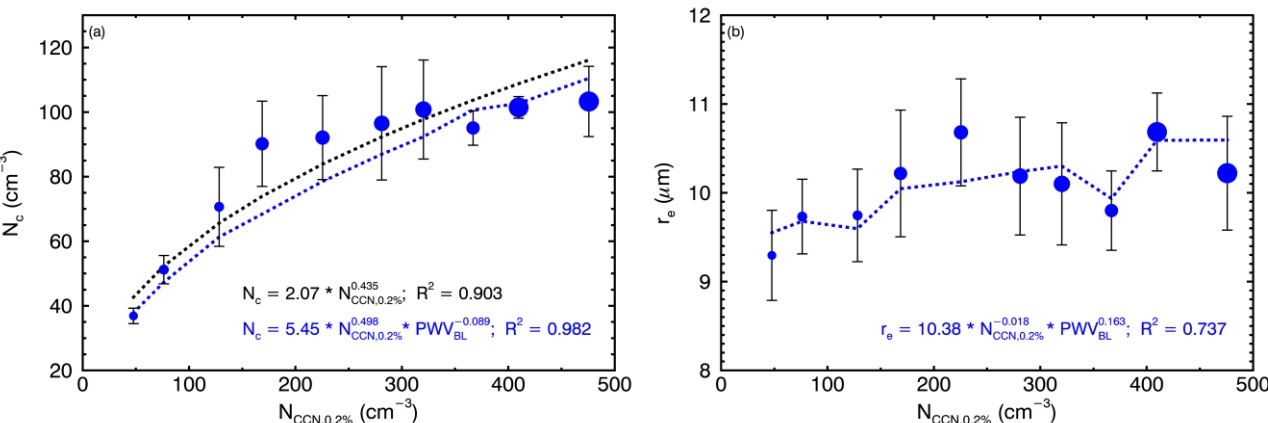

**Figure 2.** (a) $N_c$ and (b) $r_e$ as a function of $N_{CCN,0.2\%}$ (x-axis) and PWV (blue filled circles) for all selected samples. The larger blue circles represent relatively higher PWV values. Whiskers denote one standard deviation for each bin.

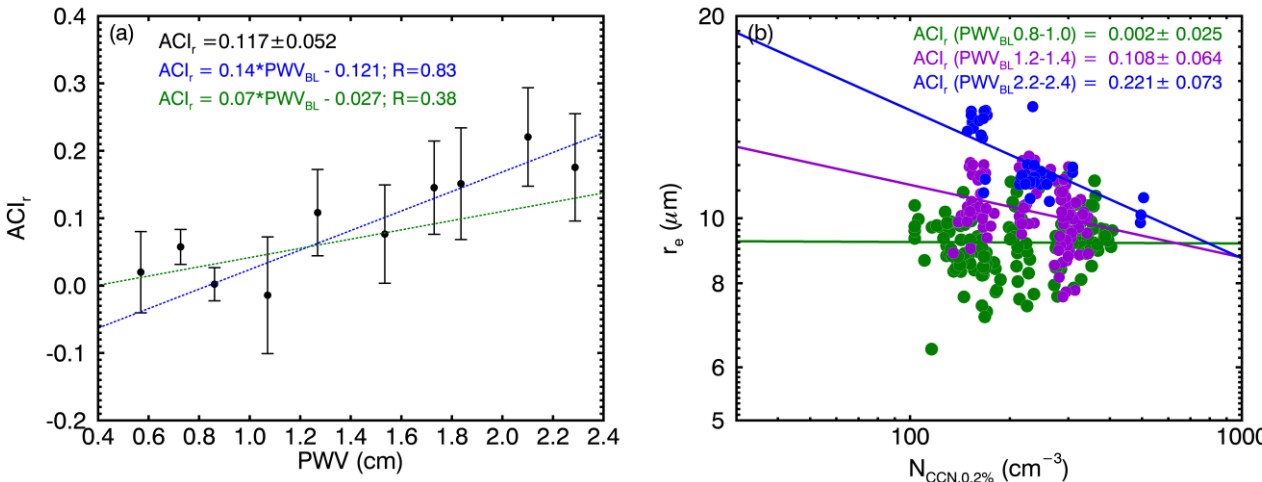

**Figure 3.** (a) Relationship of $ACI_r$ (dots) to binned $PWV_{BL}$. Whiskers denote one standard deviation for each bin. Linear regressions are performed in relatively low $PWV_{BL}$ regime (< 1.4 cm, green) and high $PWV_{BL}$ regime (> 1.4 cm); and (b) illustration of $ACI_r$ derived from $r_e$ to $N_{CCN,0.2\%}$ in following three $PWV_{BL}$ bins: 0.8-1.0 cm (green), 1.2-1.4 cm (purple), 2.2-2.4 cm (blue). The $ACI_r$ represents the relative change of $r_e$ with respect to the relative change of $N_{CCN,0.2\%}$, where positive $ACI_r$ denotes the decrease of $r_e$ with increased $N_{CCN,0.2\%}$ under binned PWV.

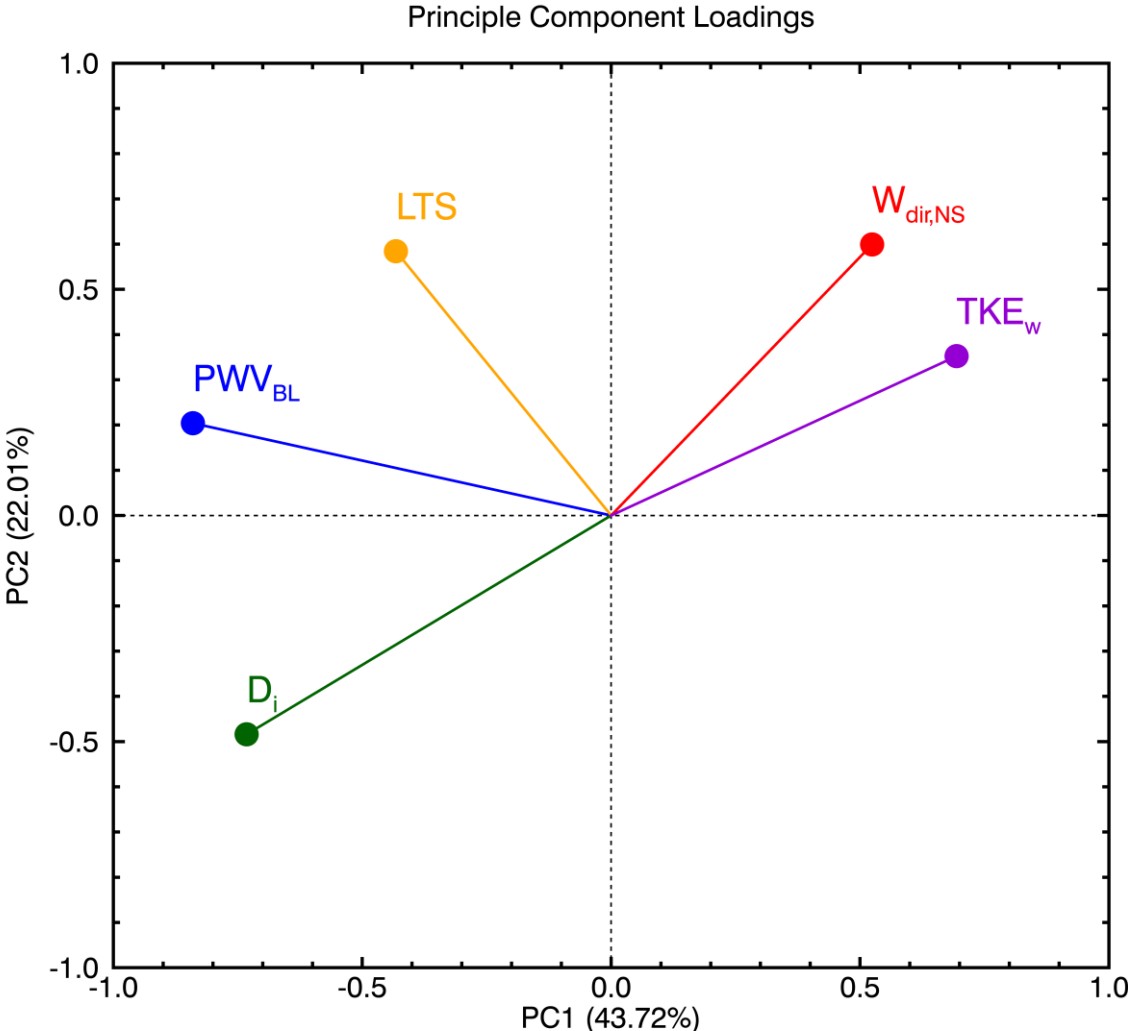

**Figure 4.** The projections of $TKE_w$ (purple), $W_{dir,NS}$ (red), LTS (orange), $PWV_{BL}$ (blue) and $D_i$ (green) onto the first principal component (PC1) and the second principal component (PC2). The x-coordinates denote variables' correlations with PC1, and the y-coordinates denote variables' correlations with PC2.

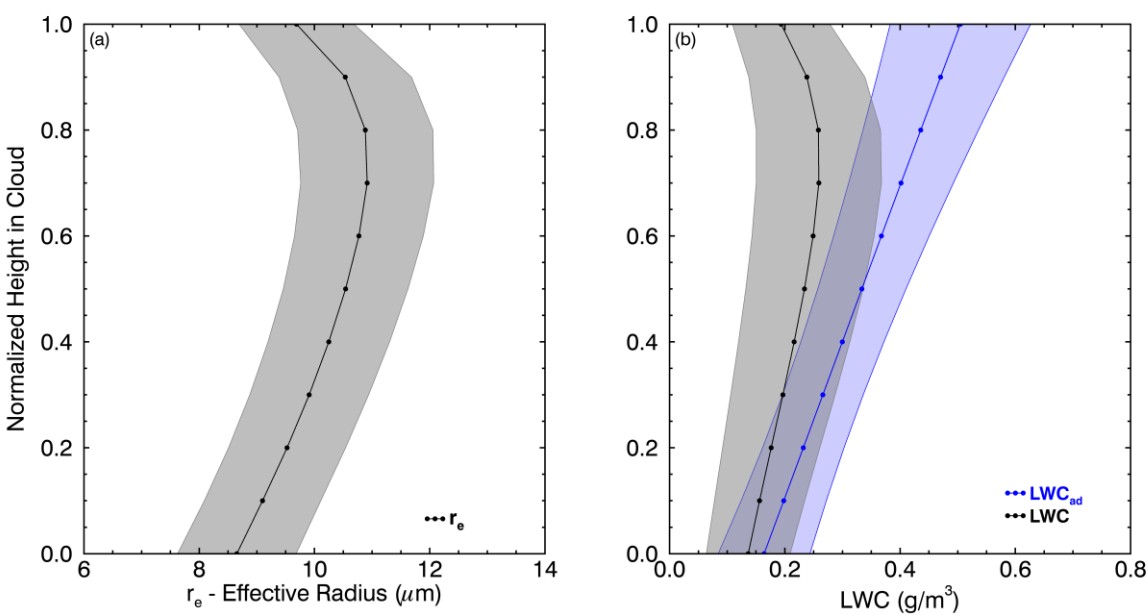

**Figure 5.** Normalized in-cloud vertical profiles of retrieved (a) $r_e$ and (b) LWC (black) and calculated adiabatic $LWC_{ad}$ (blue) for all selected cloud cases, 0 is cloud base and 1 is cloud top. Solid dotted lines denote mean values and shaded areas denote one standard deviation at each height.

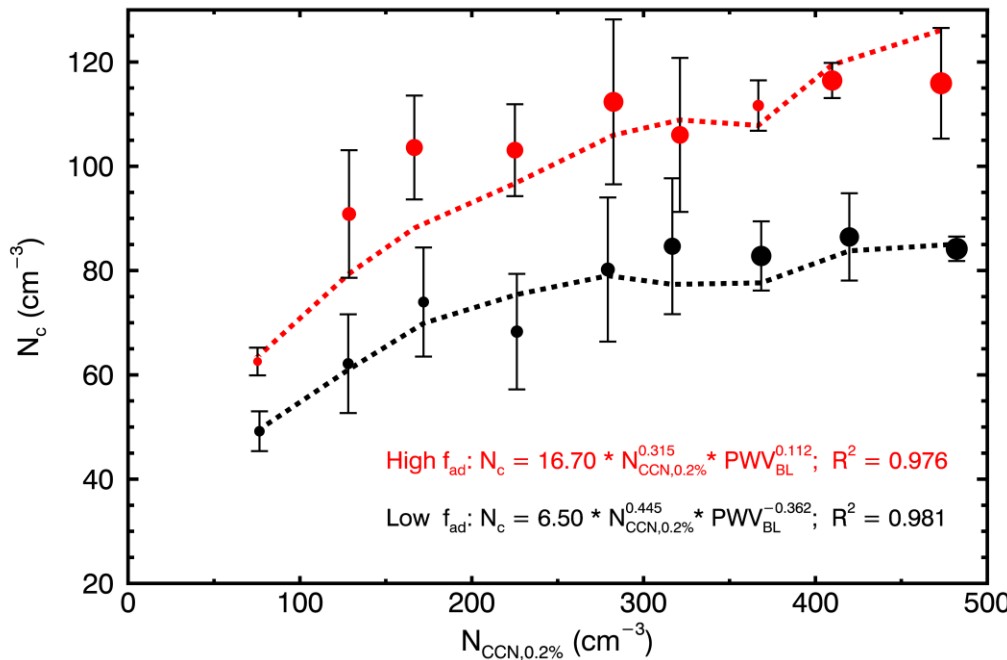

**Figure 6.** $N_c$ as a function of $N_{CCN,0.2\%}$ (x-axis) and PWV (dots) for high adiabaticity $f_{ad}$ (red) and low $f_{ad}$ (black) regimes. The larger circles represent relatively higher PWV values. Whiskers denote one standard deviation for each bin.

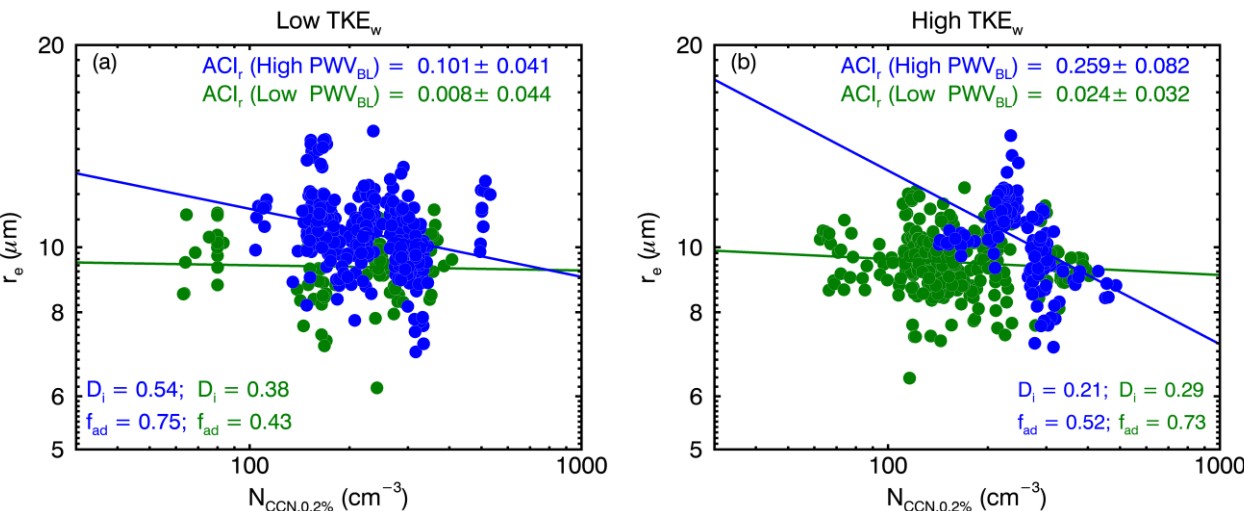

**Figure 7.** ACI$_r$ derived from $r_e$ to $N_{CCN,0.2\%}$ for (a) low TKE$_w$ and (b) high TKE$_w$ regimes. Samples in the low PWV regime are plotted in green, and samples in the high PWV regime are plotted in blue. The mean values of $D_i$ and $f_{ad}$ are displayed for each quadrant with the corresponding color-coded.

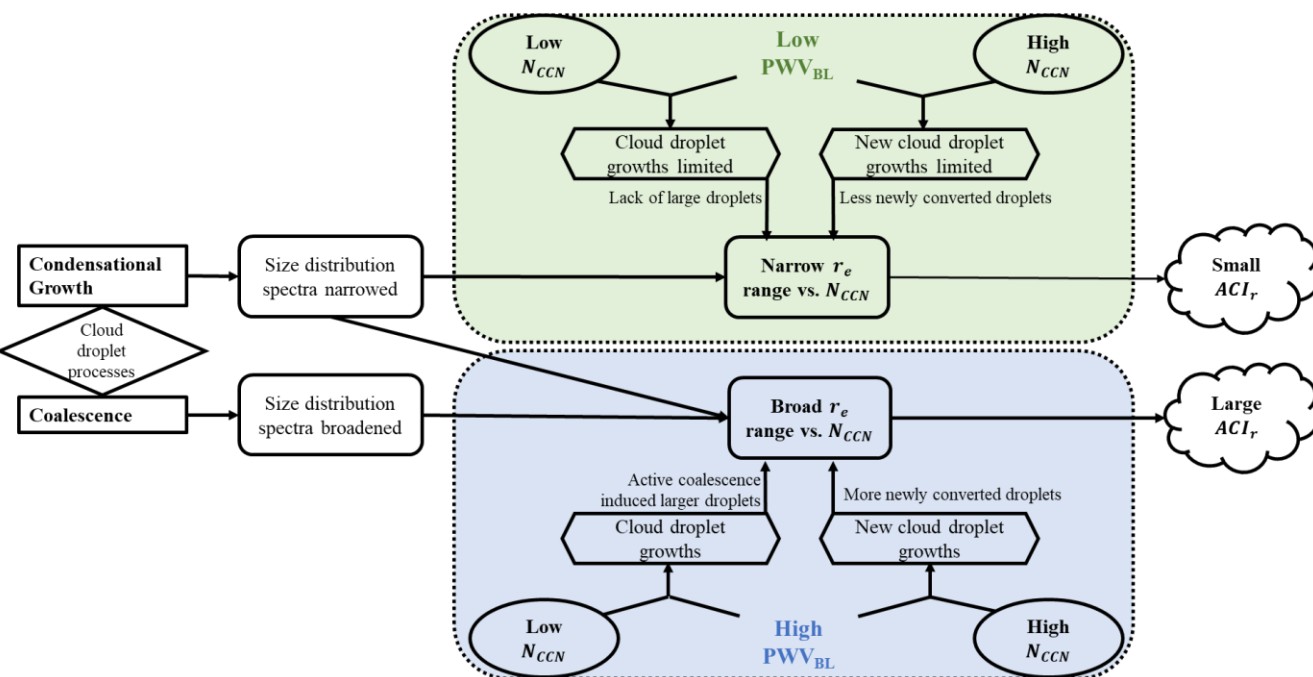

**Figure 8.** Theoretical mechanism of the responses of cloud droplet size distributions to different CCN intrusion, under relative insufficient (low PWV$_{BL}$) versus sufficient (high PWV$_{BL}$) water vapor availabilities.