# Peer review of "Environmental Effects on Aerosol-Cloud Interaction in non-precipitating MBL"

_Atmospheric Chemistry and Physics, 2021_

## Author Comment (AC1)

**Response to Reviewer #1**

We appreciate your time for carefully reviewing our manuscript. We would like to thank you for the constructive comments and suggestions, which encourage and help us to improve the manuscript. The manuscript has been revised accordingly. In the response below, your comments are provided in black text and our responses are provided in blue text.

**Response:**

Using a total of 20 non-precipitating single-layer marine boundary layer (MBL) stratus and stratocumulus cloud cases over the eastern north Atlantic (ENA) ocean, this study investigates the impacts of the environmental variables on the aerosol-cloud interaction (ACIr). Interesting results have been found with valuable discussions. For example, it shows that the ACIr values vary from -0.004 to 0.207 with increasing precipitable water vapor (PWV) conditions, indicating that re is more sensitive to the CCN loading under sufficient water vapor supply, owing to the combined effect of enhanced condensational growth and coalescence processes associated with higher cloud droplets and PWV. The paper is also well written. I would recommend its acceptance for publication after necessary minor revisions.

Detailed comments;

Line 41-44, two "verbs" exist for this sentence, which should be rephrased. Also, a few more studies are recommended here, particularly the longwave radiative property change of clouds by aerosols, such as Garrett and Zhao (2006, Doi:10.1038/nature04636).

The sentence is rephrased, and the citation is added.

Line 48-52, a few similar studies have also been carried out over the western pacific regions, which might be worthy to mention, such as Zhao et al. (2019, Doi:10.3390/atmos10010019), and Yang et al. (2019 , Doi:10.1016/j.atmosres.2019.01.027).

The citations are added.

Line 66-69, Qiu et al. (2017, Doi:10.1016/j.atmosenv.2017.06.002) showed negative relationship between cloud re and aerosol amount for low precipitable water vapor condition in spring, fall and winter at southern great plain site, but positive relationship between cloud re and aerosol amount for high precipitable water vapor condition, which could be also cited

here. Similar findings have also been found over other locations, such as western pacific region near Hebei province, China.

The citation is added.

Line 281-283, similar height normalization method has been proposed and used by Zhao et al. (2018, Doi:10.1002/2017EA000346), which is worthy to mention here. Also, Similar findings (Line 283-287) have been found earlier in several studies, including the study mentioned here.

The citation is added, in section 3.5.1 of the revised manuscript.

Line 319, Eq. (2). Earlier studies often define this for fixed LWC. How could the different definition affect the results?

The LWC/LWP describes the liquid water (i.e., existing cloud droplets), so physically linked to the $r_e$ and $N_c$. Mathematically, they have interdependent relationship in the cloud retrieval procedures, and hence to a certain extent, share the co-variabilities with the cloud microphysical properties (Dong et al., 1998; Wu et al., 2020a). In this study, by using the PWV as a sorting variable, we are trying to capture the role of ambient available water vapor in the cloud droplet growth process (especially the water vapor diffusional growth), using measurement independent to the cloud retrievals.

The discussion above is added, in section 3.3 of the revised manuscript.

Line 332-343, These are interesting findings and explainations. I wonder if this is related to the supersaturation adoped for CCN observed, or related to the true supersaturation status within clouds.

In order to investigate the theoretical implication of supersaturation conditions on the aerosol-cloud interaction observed here in the MBL stratiform clouds, the $ACI_r$ values are calculated with respect to the surface $N_{CCN}$ theoretically at two additional high supersaturation levels (0.5% and 1.2%), under all $PWV_{BL}$ conditions. The results in Table 3 show that the $ACI_r$ signals are both weak and do not have significant changes under relatively

lower PWV$_{BL}$ conditions, while the ACI$_r$ signals tend to strengthen with the increase of supersaturation under the relatively higher PWV$_{BL}$. Base on the Köhler theory, if the supersaturation exceeds the critical point for the given droplet, the droplet will thus experience continued growth, so theoretically the ACI should increase with the supersaturation under same aerosol number concentration. However, the observed limited water vapor cannot support this ideal droplet growth, results in weak responses of cloud droplets to aerosol intrusion. With the increase of observed water vapor, the continued growth of cloud droplets becomes more plausible, hence the high supersaturation yields larger droplets with low number of aerosols, more efficient droplet activation with a large number of aerosols, and in turns, larger ACI$_r$ (even out of the theoretical bounds). However, considering these high supersaturation environments are unphysical in the observed MBL cloud layers, and estimating the real supersaturation conditions using ground-based remote-sensing is beyond the scope of this study, we chose the supersaturation level of 0.2% because it represents the most typical supersaturation conditions of MBL stratiform clouds.

**Table 3.** ACI$_r$ calculated with respect to $N_{CCN}$ theoretically at different supersaturation levels, under all PWV$_{BL}$ conditions

| PWV$_{BL}$ (cm) | 0.4-0.6 | 0.6-0.8 | 0.8-1.0 | 1.0-1.2 | 1.2-1.4 | 1.4-1.6 | 1.6-1.8 | 1.8-2.0 | 2.0-2.2 | 2.2-2.4 |
|---|---|---|---|---|---|---|---|---|---|---|
| ACI$_r$ ($N_{CCN}$@0.2%SS) | 0.020 | 0.057 | 0.002 | -0.014 | 0.108 | 0.076 | 0.145 | 0.151 | 0.221 | 0.175 |
| ($N_{CCN}$@0.5%SS) | 0.023 | 0.057 | 0.0002 | 0.024 | 0.129 | 0.121 | 0.309 | 0.136 | 0.293 | 0.159 |
| ($N_{CCN}$@1.2%SS) | 0.023 | 0.045 | 0.002 | 0.072 | 0.125 | 0.123 | 0.323 | 0.175 | 0.347 | 0.186 |

The discussion above is added, in the last paragraph of section 3.3 of the revised manuscript.

Line 358-376, The mechanism proposed here is valuable. If possible, I would suggest the authors illustrate the mechanism proposed here with a diagram.

The diagram is added as Figure 8 in the revised manuscript as follows:

[Figure]

**Figure 8.** Theoretical mechanism of the responses of cloud droplet size distributions to different CCN intrusion, under relative insufficient (low PWV$_{BL}$) versus sufficient (high PWV$_{BL}$) water vapor availabilities.

Line 390, "that more close to adiabatic" shuold be "that are more close to adiabatic"

This sentence is removed in the revised manuscript.

Line 432, "to narrows the DSD" should be "to narrow the DSD"

The word 'narrows' is changed to 'narrow'.

---

## Author Comment (AC2)

**Response to Reviewer #2**

We appreciate your time for carefully reviewing our manuscript. We would like to thank you for the constructive comments and suggestions, which encourage and help us to improve the manuscript. The manuscript has been revised accordingly. In the response below, your comments are provided in black text and our responses are provided in blue text.

**Response:**

As the title suggests, this paper describes an aggregated analysis of aerosol-cloud interaction (ACI) in non-precipitating marine boundary layer clouds at the Eastern North Atlantic ARM remote sensing supersite. A relatively narrow view of ACI is taken in which the bivariate relationship between aerosol and cloud drop number concentration and the ACI index were calculated numerous times, compositing by various column-mean or column-integral quantities (e.g., water vapor path, cloud adiabaticity, lower tropospheric, turbulence). My main concern with the study is that each of these purported controlling factors is analyzed in isolation, which implicitly assumes no covariability among them. This assumption is not valid and no attempt to address this issue was given. As such, I find it difficult to accept many of the mechanistic arguments made by the authors. They cannot demonstrate cause and effect, and there are clearly confounding variables that limit their ability to draw stronger conclusions (for example, lines 243-244: "the coincidence of high NCCN and PWV does not necessarily imply a physical relationship"). I therefore recommend the manuscript be rejected and the authors encouraged to resubmit after broadening their analysis. The premise of evaluating ACI with the authors' retrieval product is promising, but to understand the role of the controlling factors, they must be analyzed in a multi-dimensional framework (principal component analysis, k-means clustering, etc.) that allows the authors to identify and, more importantly, interpret co-variability among environmental factors. As it currently stands, the conclusions of this study point vaguely toward correlations with large-scale variables but give no clear guidance.

Thanks for the constructive suggestions. To better address the reviewer's concern about the co-variabilities between the environmental variables and to more clearly shed light on their impacts on ACI, we have now conducted the principal component analysis (PCA). The variables of sub-cloud precipitable water vapor (PWV$_\text{BL}$), the boundary layer decoupling index ($D_i$), the vertical component of the turbulence kinetic energy (TKE$_\text{w}$), the lower

tropospheric stability (LTS) and the surface wind directions in terms of northerly and southerly ($W_{dir,NS}$) are constructed as the input of the eigenanalysis. Results show that the first three PCs can describe the majority (~84%) of the variance among the selected variables. Where the most explanatory PC1 (account for 43.72% contribution) strongly correlated with $PWV_{BL}$, $D_i$ (both negatively) and $TKE_w$ (positively), and hence describe the co-variation of the boundary layer conditions. While the PC2 and PC3 (account for 22.01% and 18.26% contributions, respectively) are strongly correlated with the LTS and $W_{dir,NS}$, which likely indicates the variations of the Azores High position and strength. By projecting the variables onto PC1 and PC2, the PCA loading analysis shows that the $TKE_w$ are strongly negatively correlated with $D_i$, which as expected since a more decoupled MBL is often separated into two layers where the lower one can cap the surface moisture, while the higher $TKE_w$ denote sufficient turbulence that maintains the well-mixed MBL. Additionally, the island effect is also indicated by the eigenanalysis, where the surface northerly wind would induce additional updraft velocity and hence disturb the $TKE_w$, owing to the topographic effect of the cliff north of the ENA site. Upon the PCA results, the role of cloud adiabaticities on the behaviors of CCN-$N_c$ conversion is further examined using both binning and eigenanalysis. And the factors that have the most influence on the explanatory PCs are selected as the sorting variables in the $ACI_r$ assessments.

The detailed discussions on the multi-dimensional PCA have been added to the section 3.4 of the revised manuscript as follows:

**3.4 The co-variabilities of the meteorological factors**

[revised manuscript text omitted]

In addition, the detailed results and discussions on the impacts of meteorological factors on aerosol and cloud properties, and aerosol-cloud interactions can be found in the section 3.5 of the revised manuscript.

I have a number of other concerns the authors may also wish to consider:

- How good of a proxy is PWV for PBL relative humidity? Are there cases when non-drizzling stratocumulus occur with a relatively moister free troposphere? Perhaps you could estimate the fraction of PWV in the PBL using the interpolated sonde product or Raman lidar (note: Raman will only get you subcloud vapor)?

Thanks for the comment and suggestions. In the revise manuscript, we changed to use the sub-cloud boundary-layer PWV (PWV$_{BL}$), and tested the contribution of PWV$_{BL}$ to column PWV. The discussion has been added to the section 2.2 in the revised manuscript as follows:

To capture the information of MBL water vapor more accurately, the sub-cloud boundary layer integrated precipitable water vapor (PWV$_{BL}$) is calculated using the interpolated sounding product following:

$$\mathrm{PWV_{BL}} = \frac{1}{\rho_w}\sum(z_{i+1} - z_i) * (\rho_{v,i+1} + \rho_{v,i})/2, \tag{1}$$

where the $\rho_w$ is the liquid water density and the $\rho_v$ is the water vapor density collected from the Interpolated Sounding and Gridded Sounding Value-Added Products (Toto and Jensen, 2016), the subscripts $i$ and $i + 1$ represent the bottom and top of each interpolated sounding height layer. Both PWV and PWV$_{BL}$ are temporally collocated to 5-min resolution and plotted against each other in Fig. S1a to test the contribution of PWV$_{BL}$ to the PWV. The Pearson correlation coefficient of 0.85 shows that the PWV$_{BL}$ are strongly positively correlated with the PWV, while the distribution of the percentage ratio of PWV$_{BL}$ to PWV (Fig. S1b) indicates that, on average, the PWV$_{BL}$ contribute to ~58% of the PWV. Considering the cloud-topped MBL, the majority of cases (~74%) associate with a relatively moist boundary layer compared to the amount of water vapor in the free troposphere, where the PWV$_{BL}$ already contributed over 50% of the total column PWV. In contrast, only ~9% of cloud samples occur under a relatively dry boundary layer and moist free troposphere, where PWV$_{BL}$ contributions are less than 40%. In general, the PWV can well capture the variation of the PWV$_{BL}$. In the rest of the study, the PWV$_{BL}$ are used, as it represents the sub-cloud boundary layer water vapor availabilities which are more closely related to the MBL cloud processes.

[Figure]

**Figure S1.** (a) Scatterplot of PWV versus $PWV_{BL}$; and (b) distribution of the percentage ratio of $PWV_{BL}/PWV$.

- Not enough information is given about how the vertical velocity variance TKEw is calculated. Is it a PBL average? A Doppler lidar column-deep average? Column max. value? And what Doppler lidar product are you using to get variance? The standard 10-minute integration? The median value seems low for surface-coupled stratocumulus cases. Are you evaluating any decoupled cases? There is also a diurnal and season cycle of turbulence at this site (at least, when sampling an undisturbed marine airmass; see more below), which may also be affecting your statistics.

In this study, the vertical component of the turbulence kinetic energy ($TKE_w$) are used, which is defined as:

$$TKE_w = \frac{1}{2}\overline{(w')^2} ,$$ (2)

where the $(w')^2$ is the variance of vertical velocity measured from the Doppler lidar standard 10-min integration, which collected in the Doppler Lidar Vertical Velocity Statistics Value-Added Product (Newson et al., 2019). The noise correction has been applied to reduce the uncertainty of the variance to ~10% (Hogan et al., 2009; Pearson et al., 2009). In this study, the mean value of $TKE_w$ in the sub-cloud boundary layer proportion of the Doppler lidar range is used, and the data temporal resolution is further downscaled to 5-min for temporal collocation purposes.

The description of $TKE_w$ above has been added to the section 2.2 of the revised manuscript.

We have also included the decoupling index ($D_i$) given by: $D_i = (z_b - z_{LCL})/z_b$, where the $z_{LCL}$ is the lifting condensation level calculated analytically following the method in Romp (2017), with an uncertainty of around 5 m. The surface temperature, pressure, relative humidity, and mass fraction of water vapor that used in the $z_{LCL}$ calculation, as long as the vector-averaged wind directions (in 360° coordinate) over the ENA site are obtained from the ARM surface meteorology systems (ARM MET handbook, 2011).

In this study, we are trying the examine the environmental effects on $ACI_r$ under the diverse conditions and whether the $ACI_r$ can be distinguished by them, so that we did not have prior selection on any particular environmental factors (except only the non-precipitating stratiform cloud cases), and thus the samples including strongly decoupled, moderate-to-loosely decoupled and coupled MBL conditions.

From the PCA, the $TKE_w$ has been found to be strongly positively correlated with $W_{dir,NS}$ and negatively correlated with $D_i$, which means the values of $TKE_w$ already account for the co-variabilities in these variables. Therefore, treating $TKE_w$ as the sorting variables would lead to a more physical process-orientated assessment. And the corresponding discussion is revised in section 3.5.2 of the revised manuscript.

- Have you controlled for wind direction in your analysis? It has been shown that there is an island effect when the surface wind is from the island (e.g., Zheng, Rosenfeld and Li 2016). Overland flow affects boundary layer turbulence and may also impact surface fluxes, PBL depth and CCN composition.

We have considered the potential impact of the wind direction on the boundary layer turbulence, and added to the PCA. In addition, the following summary on the island effects has been added to section 3.4 of the revised manuscript:

In Fig. 4, both $TKE_w$ and $W_{dir,NS}$ vectors are located in the same quadrant (positive in both PC1 and PC2) and close to each other with a small degree of an acute angle, which means the $TKE_w$ are strongly correlated with the $W_{dir,NS}$. When the surface wind is coming

from the north side of the island, the topographic lifting effect of the cliff would induce additional updraft over the ENA site (Zheng et al., 2016), so that the wind closer to the northerly wind (larger $W_{dir,NS}$) is more correlated with higher $\text{TKE}_w$. Therefore, the values of $\text{TKE}_w$ already account for the co-variation of $\text{TKE}_w$ and $W_{dir,NS}$.

- How much does LTS tell us at a site like ENA, and what physical motivation do you have for including it as a sorting variable? I always envision LTS as having the most meaning in the subtropical eastern boundary current (EBC) areas, i.e., northeast/southeast Pacific and southeast Atlantic. The Azores are more of a mixed subtropical/midlatitude site that has much warmer SST than in the traditional EBC areas where MBL clouds are studied, and much of the cloud cover at ENA occurs in transient postfrontal subsidence vs. longer-lasting large-scale subsidence where the spatial gradient (of both subsidence and SST) matters more in defining cloud type transitions.

We agree with your comment that the LTS might not be a feasible variable to use over ENA site, we included the LTS as it is orthogonal to the $\text{TKE}_w$ from the PCA and thus can be treated as independence. We have added the relative discussion in section 3.5.2 of the revised manuscript:

Combining LTS and $\text{PWV}_{BL}$ as sorting variables, the $\text{ACI}_r$ values for four regimes are shown in Fig. S4. The $\text{ACI}_r$ differences between low and high $\text{PWV}_{BL}$ regimes are still retained. In the low $\text{PWV}_{BL}$ regime, the $\text{ACI}_r$ values are limited to 0.016 and 0.056 for low and high LTS regimes, respectively. In the high $\text{PWV}_{BL}$ regime, the $\text{ACI}_r$ values are 0.150 and 0.171 for low and high LTS regimes, respectively, which is about 3-5 times greater than those in low $\text{PWV}_{BL}$ regime. However, the $\text{ACI}_r$ in different LTS regimes cannot be distinctly differentiated ($\text{ACI}_r$ differences between LTS regimes are ~0.02 and ~0.04), and the main difference in $\text{ACI}_r$ are still induced by the $\text{PWV}_{BL}$. Owing to the location of the ENA site where it locates near the boundary of mid-latitude and subtropical climate regimes, the MBL clouds over the ENA are found to be often under the influences of cold fronts associated with mid-latitude cyclones, where the cloud evolutions are subject to the combine effects of post-frontal and large-scale subsidence (Wood et al., 2015; Zheng et al., 2020; Wang et al., 2021). Therefore, over the ENA, although the spatial gradient of LTS is studied to be associated

with the production of MBL turbulence and the change in wind direction (Wu et al., 2017), the LTS value itself is examined to has a weak impact on the aerosol-cloud interaction from this study.

- For arguments you make about the relationship between entrainment, collision-coalescence and number concentration, it is problematic that your retrieval assumes constant Nc throughout the cloud layer. When entrainment-induced evaporation and/or collision-coalescence are active, this assumption is broken. In general, I don't understand your argument that entrainment is a sink of Nc.

The Wu et al. (2020a) retrieval works as separating the reflectivity to the contributions of cloud ($Z_c$) and drizzle, the cloud procedure assumes an initial guess of the representative layer-mean $N_c$ based on the climatology over ENA sites (Dong et al., 2014), and such allows the first guess of the vertical profile of LWC based on $N_c$ and $Z_c$, and then constrains back the $N_c$ and LWC using the LWP from MWR, finally output $r_e$ (Fig.3 in Wu et al., 2020a). Therefore, the final retrieved $N_c$ is updated to in response to the cloud microphysical processes within this time-step. From the aircraft in-situ measurements during the ACE-ENA, we used the in-situ measurement during ACE-ENA to validate the retrieval outputs and found that the observed $N_c$ profile is near-constant in middle part of the cloud, with the signal of entrainment-induced depletion near the cloud top, even in the drizzling cloud where the collision-coalescence processes are more active (Wu et al., 2020a). However, it is hard and beyond the scope of the ground-based retrieval to compare the vertical dependency of depletion rate within one time-step. Therefore, as the retrieval currently work as representing the layer-mean information from the given time-step, the preferred method in this study is to compare $N_c$ at different times, where in this case are the adiabatic versus sub-adiabatic conditions which hence yields different $N_c$ that we retrieved from the ground-based snapshot perspective. From the PCA and binning analysis, the effect of cloud adiabaticities on CCN-$N_c$ conversions may shed light on interpreting the aerosol-cloud interaction under different environmental effects.

We have added the above discussion in section 3.5.1 of the revised manuscript.

- High CCN events at ENA are not only from North America. They have also been traced to North Africa and Europe.

The corresponding sentence is changed to 'A few instances of aerosol intrusions (~3%) with higher $N_{CCN,0.2\%}$ were likely a result of continental air mass transport from North America, Europe, and Africa (Logan et al., 2014; Wang et al., 2020).'

**References.**

Wu, P., Dong, X., Xi, B., Tian, J. and Ward, D. M.: Profiles of MBL Cloud and Drizzle Microphysical Properties Retrieved From Ground-Based Observations and Validated by Aircraft In Situ Measurements Over the Azores, J. Geophys. Res. Atmos., doi:10.1029/2019JD032205, 2020a.

Zheng, Y., Rosenfeld, D. and Li, Z.: Quantifying cloud base updraft speeds of marine stratocumulus from cloud top radiative cooling, Geophys. Res. Lett., doi:10.1002/2016GL071185, 2016.

Zheng, Y., Rosenfeld, D. and Li, Z.: A More General Paradigm for Understanding the Decoupling of Stratocumulus-Topped Boundary Layers: The Importance of Horizontal Temperature Advection, Geophys. Res. Lett., doi:10.1029/2020GL087697, 2020.

---

## Author Comment (AC3)

**Response to Reviewer #3**

We appreciate your time for carefully reviewing our manuscript. We would like to thank you for the constructive comments and suggestions, which encourage and help us to improve the manuscript. The manuscript has been revised accordingly. In the response below, your comments are provided in black text and our responses are provided in blue text.

**Response:**

The authors analyze the impact of the environment on the aerosol-cloud interactions (ACI) from ground observations over the eastern north Atlantic. They find that both lower-trosospheric stability and turbulent kinetic energy influence the connection between water vapor, cloud-microphysics, and subsiquently ACI. For instance, they find that higher lower-tropospheric stability leads to higher cloud drop concentrations and ACI.

Overall, I think this paper is both well thoughout and written. However, I do have a number of issues that I would appreciate clarification on. Note that, even though I split my comments between major and minor, this is more of just a distinction between general and technical comments. Therefore, I recommend publication once these comments are addressed.

Major:

Line 147: Is LTS the most appropriate variable to use over the northeast Atlantic, considering the much larger influence of midlatitude cyclones compared to subtropical regions?

We agree with your comment and the other reviewer's comment that the LTS might not be a feasible variable to use over ENA site, and thus we have added the relative discussion in section 3.5.2 of the revised manuscript:

Combining LTS and $PWV_{BL}$ as sorting variables, the $ACI_r$ values for four regimes are shown in Fig. S4. The $ACI_r$ differences between low and high $PWV_{BL}$ regimes are still retained. In the low $PWV_{BL}$ regime, the $ACI_r$ values are limited to 0.016 and 0.056 for low and high LTS regimes, respectively. In the high $PWV_{BL}$ regime, the $ACI_r$ values are 0.150 and 0.171 for low and high LTS regimes, respectively, which is about 3-5 times greater than those in low $PWV_{BL}$ regime. However, the $ACI_r$ in different LTS regimes cannot be distinctly differentiated ($ACI_r$ differences between LTS regimes are ~0.02 and ~0.04), and the main difference in $ACI_r$ are still induced by the $PWV_{BL}$. Owing to the location of the ENA site where it locates near the

boundary of mid-latitude and subtropical climate regimes, the MBL clouds over the ENA are found to be often under the influences of cold fronts associated with mid-latitude cyclones, where the cloud evolutions are subject to the combine effects of post-frontal and large-scale subsidence (Wood et al., 2015; Zheng et al., 2020; Wang et al., 2021). Therefore, over the ENA, although the spatial gradient of LTS is studied to be associated with the production of MBL turbulence and the change in wind direction (Wu et al., 2017), the LTS value itself is examined to has a weak impact on the aerosol-cloud interaction from this study.

Line 171: How many potential non-precipitating cloud cases were there, and do your results suggest that most MBL clouds produce precip over the northeast Atlantic?

During the study period we found 20 valid non-precipitating single-layer low cloud case that fit in our criteria and also lasting at least longer than 2 hours. And yes, our results support the previous study that over the ENA site, the annual mean drizzle frequency is 55%, with 70% in winter and 45% in summer (Wu et al., 2020).

Line 193: You could highlight that the median LTS of 19.1 K is close to the value (18.55 K) used by prior studies to separate stratocumulus from shallow cumulus.

The result from prior study is highlighted as follows:

Note that the median LTS of 19.1 K in this study is close to the separation threshold of 18.55K suggested by prior studies to distinguish the marine stratocumulus from a global assessment of marine shallow cumulus clouds (Smalley and Rapp, 2020).

Line 226: You compare the logarithmic ratio that you find to other studies, but I don't understand what it actually means.

The ratio reflects the relative conversion efficiency of cloud droplets from the CCN, regardless of the water vapor availabilities. Theoretically it has the boundaries of 0 - 1, where the lower bound means no change of $N_c$ with $N_{CCN}$, and the upper bound indicates a linear relationship that every CCN would result in one cloud droplet. Our result is comparable with the previous studies that also targeting the MBL stratiform clouds, indicates a certain similarity of the bulk

cloud microphysical responses with respect to aerosol intrusion in those type of cloud and over different marine environments, further support that the assessment in this study is valid.

The discussion above is added in section 3.2 in the revised manuscript.

Figures 5 - 7: There doesn't appear to be much of a trend in the scatter plots, so what is the $R^2$ value for these regressions? Maybe this could be fixed by constraining your axes to closer to the limits of your datapoints?

In the revise manuscript, we changed to use the sub-cloud $PWV_{BL}$ in sorting the data, as suggested by reviewer #2. We have constrained the plotting axes to be closer to the data points. Since the values of $ACI_r$ have a theoretical upper bound of 0.33 (McComiskey et al., 2009), so even the largest $ACI_r$ will probably not showing a steep trend in the scatterplot. However, the slopes of regression can be distinguished, all linear regressions for those groups of data have been tested by two-tailed T statistic and pass the 95% significant level.

Minor:

Line 77: "relatively shallower" should be "relatively shallow"

The word is changed to 'shallow'.

Line 78: I think "and is prone to" should be and "are prone to"

It is changed to 'are prone to'.

Line 80: "marine boundary layer maintained by" should be "marine boundary layer which is maintained by"

The 'which is' is added to the sentence.

line 85: "regime of active coalescence process" should either be "regime of the active coalescence process" or "regime of active coalescence"

It is changed to 'regime of active coalescence'.

line 106: "particularly disentangling" should be "particularly by disentangling"

It is changed to 'particularly by disentangling'.

line 121: "operates at 910 nm laser beam" doesn't make sense, and maybe could be "operates at 910nm"

It is changed to 'operates at 910 nm'.

line 159: "from Doppler lidar" should either be "from a Dopplar lidar" or "from the Dopplar lidar"

It is changed to 'from the Doppler lidar'.

line 183: "lay" should be "lie"

It is changed to 'lie'.

line 388: Unless I missed something why is Figure 5b discussed before Figure 5a, could you just flip those subpanels?

The figure subpanel is flipped.

Figure 1: This may just be my printout, but the median dashed lines are difficult to see. Could you use a thicker line or a different color?

The median value is now displayed directly on each subpanel in Figure 1.

**Reference:**

McComiskey, A, Feingold, G., Frisch, A. S., Turner, D. D., Miller, M., Chiu, J. C., Min, Q. and Ogren, J.: An assessment of aerosol-cloud interactions in marine stratus clouds based on surface remote sensing, J. Geophys. Res., 114, D09203, doi:10.1029/2008JD011006, 2009.

Smalley, K. M. and Rapp, A. D.: The role of cloud size and environmental moisture in shallow cumulus precipitation, J. Appl. Meteorol. Climatol., doi:10.1175/JAMC-D-19-0145.1, 2020.

Wu, P., Dong, X. and Xi, B.: A climatology of marine boundary layer cloud and drizzle properties derived from ground-based observations over the azores, J. Clim., doi:10.1175/JCLI-D-20-0272.1, 2020.

---

## Referee Report (RR1)

Overall, the addition of the PCA results greatly strengthens the results of the manuscript and I find the significant revisions by the authors have ameliorated the gravest of my earlier concerns. Despite these greatly improved state of the study, I have several minor technical comments and suggested typographical edits that I would like to see addressed to clarify language in the paper before it is accepted. Many of the extended new passages seem rather hastily worded. I recommend the authors have a native English speaker or editing service review newly incorporated text.

MINOR COMMENTS (L refers to line number)
L31-32: What do you mean by the phrase "underneath CCN and moisture sources?" Do you mean surface sources? Or sources underneath the cloud that aren't the surface?

L53-55: You need to rephrase the sentence beginning "The enhance Nc conversion…" as it is very difficult to understand. When you say "Nc conversion" do you mean activation of CCN or autoconversion? In addition, what do you mean by "intrusions of CCN?" Intrusions from where? Advected aerosol, stronger sources, from the free troposphere?

L180: Do you mean northeast Atlantic instead of southeast?

Section 3.5.1: Can you spell out the implications of the moderate negative correlations between cloud properties and PC1? In my mind, greater TKE should lead to a more adiabatic PBL cloud so it is worth explicitly pointing out that adiabaticity (probably) depends more on decoupling state than turbulence. This is a somewhat counter-intuitive argument, maybe a weakness of your use of sub-cloud TKE vs. full PBL mean including the cloud layer. In addition, I think the low correlations you find between PC2 and cloud properties are likely a consequence of subsetting for single-layer stratocumulus. If you looked at *all* boundary layer clouds at ENA, you would naturally be sorting by LTS and wind direction.

General comment on sections 3.4-3.5: I wonder if the missing factor that could explain the D – TKE negative correlation is surface fluxes? There is likely a diurnal cycle factor as well, perhaps obtainable through examining net cloud top longwave cooling rate

TYPOGRAPHICAL COMMENTS
L129: Instead of "where sits in," try "which sits in"

L130-132: This sentence is rather colloquial in tone. I suggest removing "So that" at the beginning.

L184: singular, "buoyancy generation and shear"

L186: "growth process" instead of "growing process"

L972-977: this is a mega-long sentence and is rather awkwardly worded. Please revisit to simplify and clarify wording.

L1061: "to have" instead of "to has"

---

## Author Response (AR2)

**The author's responses to the reviewer's comments and the revised manuscript are merged below.**

**Response to Reviewer #2**

Overall, the addition of the PCA results greatly strengthens the results of the manuscript and I find the significant revisions by the authors have ameliorated the gravest of my earlier concerns. Despite these greatly improved state of the study, I have several minor technical comments and suggested typographical edits that I would like to see addressed to clarify language in the paper before it is accepted. Many of the extended new passages seem rather hastily worded. I recommend the authors have a native English speaker or editing service review newly incorporated text.

We appreciate your time for carefully reviewing our manuscript. We would like to thank you for the constructive comments and suggestions, which encourage and help us to improve the manuscript. The manuscript has been revised accordingly, and Dr. Timothy Logan has done a thorough grammar check for the manuscript. In the response below, your comments are provided in black text and our responses are provided in blue text.

**Response:**

MINOR COMMENTS (L refers to line number)

L31-32: What do you mean by the phrase "underneath CCN and moisture sources?" Do you mean surface sources? Or sources underneath the cloud that aren't the surface?

This phrase is changed to '…and the below-cloud CCN and moisture sources' in the revised manuscript.

L53-55: You need to rephrase the sentence beginning "The enhance Nc conversion…" as it is very difficult to understand. When you say "Nc conversion" do you mean activation of CCN or autoconversion? In addition, what do you mean by "intrusions of CCN?" Intrusions from where? Advected aerosol, stronger sources, from the free troposphere?

This phrase is changed to '…enhanced activation of CCN and the cloud droplet condensational growth induced by the higher below-cloud CCN loading' in the revised manuscript.

L180: Do you mean northeast Atlantic instead of southeast?

Thanks for capturing. This term is changed to 'northeast Atlantic' in the revised manuscript.

Section 3.5.1: Can you spell out the implications of the moderate negative correlations between cloud properties and PC1? In my mind, greater TKE should lead to a more adiabatic PBL cloud so it is worth explicitly pointing out that adiabaticity (probably) depends more on decoupling state than turbulence. This is a somewhat counter-intuitive argument, maybe a weakness of your use of sub-cloud TKE vs. full PBL mean including the cloud layer. In addition, I think the low correlations you find between PC2 and cloud properties are likely a consequence of subsetting for single-layer stratocumulus. If you looked at *all* boundary layer clouds at ENA, you would naturally be sorting by LTS and wind direction.

Thanks for the comments and suggestions, we have expanded the discussion in the revised section 3.5.1 as follows:

These negative correlations suggest that under the higher $PWV_{BL}$ condition, the sufficient water vapor supply allows more CCN to become cloud droplets, as previously discussed, and hence increases the cloud adiabaticity due to the dominant condensational growth process. While in the situation of relatively higher $TKE_w$, the decrease in the $N_c$ and $f_{ad}$ might be partly attributed to the association with the active in-cloud coalescence process and entrainment of dry air. However, owing to the obstacle of retrieving in-cloud $TKE_w$ from the ground-based remote sensing, the usage of sub-cloud $TKE_w$ in this study captures part of the relationship between turbulence and adiabaticity. Therefore, in this situation, the cloud adiabaticity might depend more on $PWV_{BL}$ and the boundary layer decoupling state.

Additionally, the following statement is also added:

These weak correlations might likely be due to the subset of MBL single-layer stratocumulus in this study, as the previous study over the ENA found that the sensitivity of MBL cloud adiabaticity largely depends on the strength of cloud top inversion (which can be partially indicated by the increased LTS) and slightly depends on the boundary layer decoupling (Terai et al., 2019; Zheng et al., 2020).

General comment on sections 3.4-3.5: I wonder if the missing factor that could explain the D–TKE negative correlation is surface fluxes? There is likely a diurnal cycle factor as well, perhaps obtainable through examining net cloud top longwave cooling rate

Thanks for the comments, we have added the discussion in the revised section 3.4 as follows:

Note that the negative correlation between $D_i$ and $TKE_w$ examined here might also be partly attributed to the diurnal cycle of the turbulence, which is studied to be associated with the cloud-top longwave radiative cooling over the ENA, especially for the drizzling clouds (Ghate et al., 2021; Zheng et al., 2016). However, this study focuses on the non-precipitating clouds where the effect of drizzle on the cloud-top radiative cooling driven turbulence is minimum, and examining the cloud-top radiative cooling rate from ground-based remote sensing is beyond the scope of the current study. It would be with interest to get the accurate cloud-top radiative cooling rate using a radiative transfer model to perform further study in the future.

TYPOGRAPHICAL COMMENTS

L129: Instead of "where sits in," try "which sits in"

It is changed to 'which sits in' in the revised manuscript.

L130-132: This sentence is rather colloquial in tone. I suggest removing "So that" at the beginning.

The term 'So that…' is removed.

L184: singular, "buoyancy generation and shear"

It is changed to 'buoyancy generation and shear' in the revised manuscript.

L186: "growth process" instead of "growing process"

It is changed to 'growth process' in the revised manuscript.

L972-977: this is a mega-long sentence and is rather awkwardly worded. Please revisit to simplify and clarify wording.

Thanks for pointing it out. This sentence is changed to 'From the aircraft in-situ measurements during the ACE-ENA, we found that the observed vertical profile of $N_c$ is near-constant in middle part of the cloud (even in the drizzling cloud where the collision-coalescence processes are more active), and the signal of entrainment-induced $N_c$ depletion is shown near the cloud top (Wu et al., 2020a)' in the revised manuscript.

L1061: "to have" instead of "to has"

It is changed to 'to have' in the revised manuscript.

**References.**

[revised manuscript text omitted]
 $\text{ACI}_r$ under different $\text{PWV}_{\text{BL}}$ bins, and illustrates the calculation of $\text{ACI}_r$ in three different $\text{PWV}_{\text{BL}}$ ranges. Note that in Fig. 3a, the regressions are derived from all points (statistically significant with a confidence level of 95%). As shown in Fig. 3a, the $\text{ACI}_r$ values range from close-to-zero values (-0.01) to 0.22, with the mean value of $0.117 \pm 0.052$. The $\text{ACI}_r$ range of this study agrees well with the previous studies of MBL cloud aerosol-cloud interactions (McComiskey et al., 2009; Pandithurai et al., 2009; Liu et al., 2016). It is noteworthy that the variation of $\text{ACI}_r$ with $\text{PWV}_{\text{BL}}$ suggests two different relationships under separated $\text{PWV}_{\text{
[revised manuscript text omitted]